# Celf4 controls mRNA translation underlying synaptic development in the prenatal mammalian neocortex

Iva Salamon[1,2], Yongkyu Park [1], Terezija Miškić[3], Janja Kopić[3], Paul Matteson[4], Nicholas F. Page[1], Alfonso Roque[1], Geoffrey W. McAuliffe[1], John Favate [5], Marta Garcia-Forn[6,7,8,9,10], Premal Shah [5], Miloš Judaš[3], James H. Millonig[4], Ivica Kostović[3], Silvia De Rubeis [6,7,8,9,10], Ronald P. Hart [11], Željka Krsnik [3,12] ✉ & Mladen-Roko Rasin[1,12] ✉

Abnormalities in neocortical and synaptic development are linked to neurodevelopmental disorders. However, the molecular and cellular mechanisms governing initial synapse formation in the prenatal neocortex remain poorly understood. Using polysome profiling coupled with snRNAseq on human cortical samples at various fetal phases, we identify human mRNAs, including those encoding synaptic proteins, with finely controlled translation in distinct cell populations of developing frontal neocortices. Examination of murine and human neocortex reveals that the RNA binding protein and translational regulator, CELF4, is expressed in compartments enriched in initial synaptogenesis: the marginal zone and the subplate. We also find that Celf4/CELF4-target mRNAs are encoded by risk genes for adverse neurodevelopmental outcomes translating into synaptic proteins. Surprisingly, deleting *Celf4* in the forebrain disrupts the balance of subplate synapses in a sex-specific fashion. This highlights the significance of RNA binding proteins and mRNA translation in evolutionarily advanced synaptic development, potentially contributing to sex differences.

The neocortex is evolutionarily the most advanced region of the central nervous system where the higher perceptive and cognitive abilities reside[1–3]. The basic architecture of adult neocortex is almost entirely defined during developmental periods before birth; however, there are evident structural differences between prenatal and adult neocortex in all mammals, including human and mouse. Although the prenatal neocortical development consists of tightly controlled series of molecular events guiding ordered yet overlapping key cellular events (proliferation, cellular migration, lamination, cellular and laminar specification, connectivity, and initial synaptogenesis), the molecular

[1]Department of Neuroscience and Cell Biology, Rutgers University, Robert Wood Johnson Medical School, Piscataway, NJ 08854, USA. [2]Rutgers University, School of Graduate Studies, New Brunswick, NJ 08854, USA. [3]Croatian Institute for Brain Research, Center of Research Excellence for Basic, Clinical and Translational Neuroscience, University of Zagreb, School of Medicine, Zagreb 10000, Croatia. [4]Center for Advanced Biotechnology and Medicine, Department of Neuroscience and Cell Biology, Rutgers Robert Wood Johnson Medical School, Piscataway, NJ, USA. [5]Department of Genetics, Rutgers University, Piscataway, NJ 08854, USA. [6]Seaver Autism Center for Research and Treatment, Icahn School of Medicine at Mount Sinai, New York, NY 10029, USA. [7]Department of Psychiatry, Icahn School of Medicine at Mount Sinai, New York, NY 10029, USA. [8]Friedman Brain Institute, Icahn School of Medicine at Mount Sinai, New York, NY 10029, USA. [9]Mindich Child Health and Development Institute, Icahn School of Medicine at Mount Sinai, New York, NY 10029, USA. [10]The Alper Center for Neural Development and Regeneration, Icahn School of Medicine at Mount Sinai, New York, NY 10029, USA. [11]Department of Cell Biology and Neuroscience, Rutgers, The State University of New Jersey, Piscataway, NJ 08854, USA. [12]These authors jointly supervised this work: Željka Krsnik, Mladen-Roko Rasin. ✉e-mail: zeljka.krsnik@hiim.hr; roko.rasin@rutgers.edu

dynamics responsible for regulating each of these cellular events are still not completely understood.

This is particularly true for the formation of the initial synapses, which develop in the prenatal neocortical subplate (SP) layer[4–7]. In humans, the SP is the largest transient neocortical compartment, playing an essential role in the maturation of later circuits subserving sensory processing, cognition, and social behavior. It is also responsible for the earliest higher level receptive functions for sensory stimuli and language[6,8]. During development, the SP provides an interactive setting for neuronal migration, neuronal and glial differentiation and specification, dendritic growth, pruning and cell death, neurochemical maturation, myelination, axonal outgrowth and ingrowth, and synaptogenesis[4–6,9]. During early phases of human and mouse development [11–18 post-conceptual weeks (PCW); embryonic day 15 (E15) – postnatal day 0 (P0), respectively], the SP neurons begin to form the extensive synaptic network below the cortical plate (CP). This network consists of local circuitry between GABAergic and glutamatergic neurons, subcortical afferents from the thalamus (glutamatergic), basal forebrain (cholinergic), and brain stem (serotonergic, dopaminergic, noradrenergic), as well as prospective input from the developing cortical plate (both glutamatergic and GABAergic)[4,6,10–14]. While these synapses showcase structural characteristics similar to chemical synapses, their functional properties have yet to be clearly defined. Furthermore, early spontaneous neuronal coupling can also exhibit electrical properties[12]. The dynamic prenatal restructuring and specialization of SP neurons and their synaptic development guides the long-term organization of cortical networks. Indeed, a disruption of the SP has been shown to affect both the structure and functions of the neocortex during postnatal life[15,16]. These findings have collectively spotlighted the SP to understand its normal and abnormal development, and potential role in the pathogenesis of human neurodevelopmental disorders (NDDs), including autism spectrum disorder (ASD)[10–12,17]. Despite recent studies focusing on the SP, we still know very little about the molecular drivers of its formation, and prenatal synaptic development overall.

Precise orchestration of spatiotemporal changes of cellular events that shape the neocortex, including the SP, requires the regulation of gene expression at both transcriptional and post-transcriptional levels[18–24]. RNA binding proteins (RBPs), including those from the ELAV/CELF family, are major regulators of post-transcriptional mRNA fate (such as translation or protein synthesis) in the developing neocortex[2,25,26]. The ever-increasing roles of RBPs in RNA biology make them ideal drivers for accelerating the development and human-specific evolution of mRNA and protein diversity[2,3,27–31]. Consistently, genetic variation in genes encoding RBPs and other translational dysregulation, contribute to the risk for NDDs, including human specific ASD[32–35], and proteins encoded by NDD risk genes have been shown to regulate the mRNAs of the other NDD risk genes[36–38]. While substantial efforts have gone into the transcriptional profiling of neural populations across development, the impact of post-transcriptional events, including the dynamics of translational regulations by RBPs, in the prenatal synaptic development are poorly known.

By generating a comprehensive single-nucleus RNA sequencing (snRNAseq) map across early fetal and midfetal human neocortical development, we identified and tracked the development of frontal neocortical cell populations, including the SP neurons. The CUGBP Elav-Like Family Member 4 (CELF4), encoded by a risk gene for ASD and related NDDs, emerged as the top-ranked RBP associated with synaptic functions that also showed high expression in the developing SP. We further found that human CELF4 and murine Celf4 proteins translationally repress a shared group of mRNAs encoding synaptic proteins, thus contributing to synaptic development in the prenatal neocortex. Remarkably, murine Celf4 has sex-specific roles in synaptic SP development. Overall, our findings provide a deeper understanding of the spatiotemporal dynamics of prenatal human cortical development and pinpoint a novel mechanism of post-transcriptional regulation guiding it.

## Results

### Single-nucleus transcriptomic atlas of human fetal neocortical development

To capture the gene expression dynamics in the developing human frontal cortex, we sampled fetal neocortices and performed single-nucleus RNA sequencing (snRNAseq) during early fetal (11 and 12 postconceptional weeks; 11/12 PCW), early midfetal (14/15 PCW) and late midfetal (17/18 PCW) phases (Fig. 1a, Supplementary Data 1). Rather than integrating all the data into one analysis, we opted for phase-specific integration and clustering. This allowed us to accurately group nuclei into clusters matching discernible cell types, providing a more precise understanding of the developmental process. Using Uniform Manifold Approximation and Projection (UMAP) to visualize the shared nearest neighbor (SNN) clustering of the phase-specific datasets, we identified 19 early, 21 mid, and 21 late clusters (Supplementary Fig. 1a, bottom). In contrast, integrating all samples followed by optimal clustering led to the formation of clusters that do not clearly match individual cell types (Supplementary Fig. 1a, top) since some nuclei from phase-specific clusters were allocated to different integrated clusters, potentially rendering them less biologically interpretable (Supplementary Fig. 1b). This suggests that the integration approach is confounded by similar cells from other phases, therefore differing from the phase-specific clustering approach, which highlights the importance of carefully considering the most appropriate analysis strategy for the specific research question at hand. Then, we utilized previously reported cell-specific RNAs[22,39–41] to catalog distinct cell types that populate the developing neocortex and traced them across the three fetal phases. These cell clusters include SP neurons; apical radial glia (aRG) and basal radial glia (bRG); Cajal-Retzius cells (CR cells); layer 2–4 excitatory neurons (ExN L2-4) and layer 5/6 excitatory neurons (ExN L5-6); differentiating excitatory neurons (ExN_diff); oligodendrocyte precursor cells (OPC) and astrocytes (Astro); medial ganglionic eminence (MGE)-derived interneurons (MGE InN) and caudal ganglionic eminence (CGE)-derived interneurons (CGE InN); endothelial cells (Endo), pericytes (Peri), and microglia (Micro).

Distinct SP clusters emerged within each examined fetal phase and their transcriptional signatures evolved across phases, in line with the transient expansion of SP sublayers during human cortical development[42]. During early fetal period (11/12 PCW), two neuronal populations appear after the preplate split: the first population ends above the CP, in the marginal zone (MZ; cluster 16: "CR cells & SP & ExN L5/6"), while the second one ends below the CP as the deep presubplate sublayer (clusters 0, 9 and 7: "SP & ExN L5/6"). The early midfetal (14/15 PCW) period is composed of two SP clusters below the CP, called SP upper and SP lower based on the transcriptomic, morphological and functional characteristics (clusters 5 and 13: "SP"). In the late midfetal period (17/18 PCW), three SP populations (superficial SP, intermediate SP and deep SP) are expected based on morphological and functional features[9], but they appear as undistinguishable based on their transcriptional profile (clusters 4, 15 and 16: "SP"). This discrepancy between morphological/functional and transcriptomic findings at 17/18 PCW opens a possibility that post-transcriptional mechanism plays a role in later phases of development.

Genetic evidence[43,44] and transcriptomic analyses on postmortem brains sampled from unaffected individuals[22,45,46] or affected individuals contrasted with controls[47–51] have identified the midfetal neocortex as a major risk nexus for ASD. Evidence in mouse models corroborate that the SP might be altered in NDDs and ASD[52,53]. To interrogate the relationship between neocortical development and

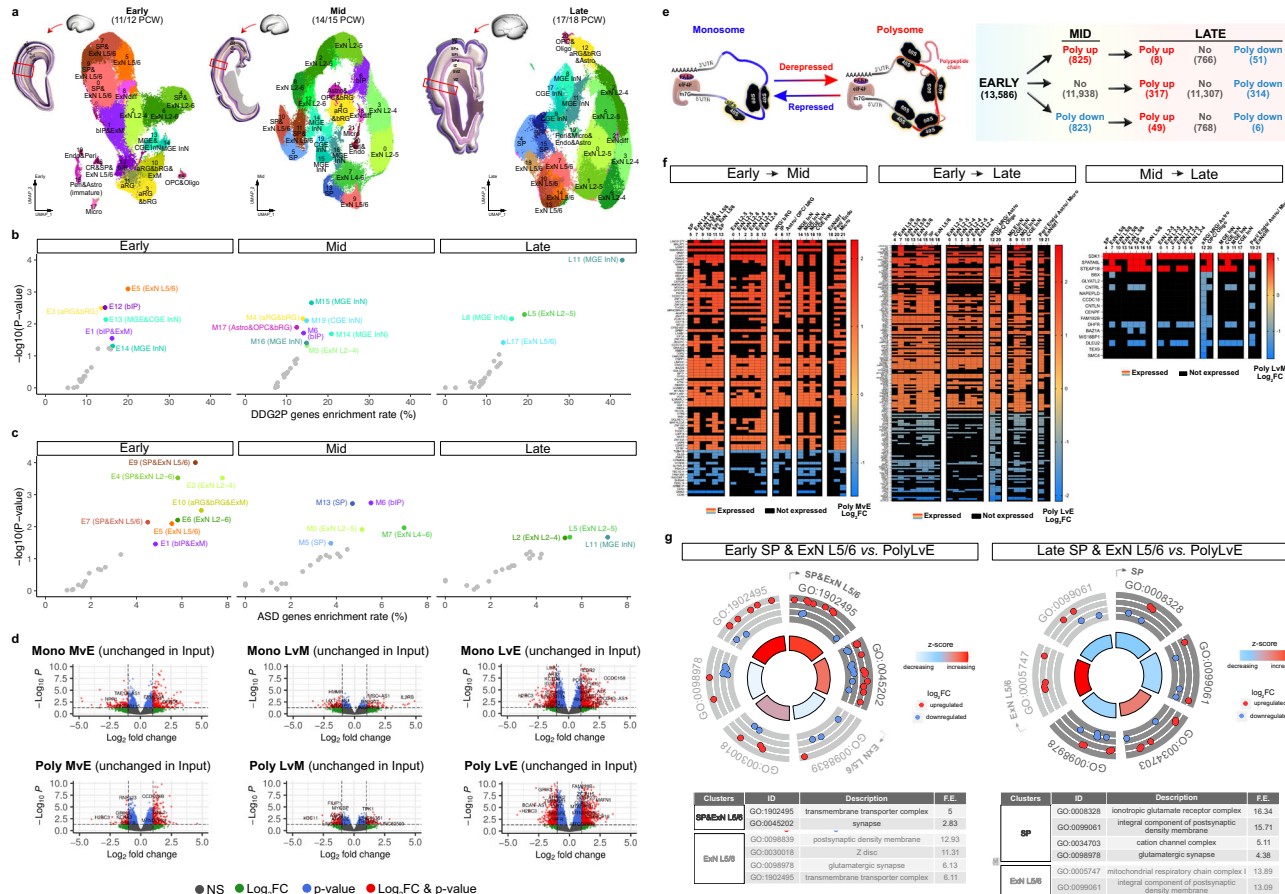

**Fig. 1 | Single-nucleus transcriptome and translational landscapes reveal cell type heterogeneity across human fetal neocortical development. a** UMAP projection of human neocortical snRNAseq data uncovered 19 cell clusters during early fetal (11/12 PCWs), 21 clusters during early midfetal (14/15 PCWs) and 21 clusters during late midfetal (17/18 PCWs) development (sample size: $n = 2$ per each phase). The most enriched genes were used to assign cell identity to each cluster: SP subplate neurons, aRG apical radial glia, bRG basal radial glia, CR Cajal-Retzius cells. ExN L2-4, layer 2–4 excitatory neurons. ExN L5-6, layer 5/6 excitatory neurons. ExN$_{diff}$, differentiating excitatory neurons. OPC, oligodendrocyte precursor cells. Astro, astrocytes. MGE InN, medial ganglionic eminence-derived interneurons (MGE InN). CGE InN, caudal ganglionic eminence-derived interneurons. Endo, endothelial cells. Peri, pericytes. Micro, microglia. **b** Enrichment analysis of genes preferentially expressed in each cell cluster shown in (**a**) (average log2FC > 0.5849) with NDD risk genes listed in DDG2P or (**c**) ASD risk genes (from;[43] see "Methods"). The plots show the empirical *P* value of the enrichment for each cluster in each developmental phase as a function of the enrichment rate (see "Methods" for statistics; Supplementary data 2). Colored/ labeled (as in **a**), clusters with significant enrichment. Gray, clusters with non-significant enrichment. **d** Volcano plots of differentially expressed mRNAs from polysome profiling-RNAseq ($n = 2$ neocortices per developmental phase) based on their distribution in monosomes (Mono) and polysomes (Poly) in early fetal *vs.* early midfetal (MvE), early midfetal *vs.* late midfetal (LvM), and early fetal *vs.* late midfetal (LvE) comparisons. Red dots, mRNAs with no changes in input (transcriptionally stable levels) but differential association with Mono (top) and Poly (bottom). Gray dots, mRNAs with unchanged levels in Mono or Poly that do not reach significance at |log2FC| >1 and adjusted *p* value < 0.05 (threshold marked as

horizontal dashed line). *P* values shown in the plot were estimated by Wald test with multiple comparison correction by the Benjamini-Hochberg method. **e** Left: mRNAs categorized as derepressed (more translationally active, right) or repressed (less translationally active, left) based on their higher association with Poly or Mono fractions, respectively. Right: Table summarizes the translational status of 13,586 mRNAs associated with actively translating polysomes in the early phase (starting point), noting their changes in later phases: increased (Poly up), decreased (Poly down), or unchanged (No). **f** Expression heatmaps of translationally changed mRNAs from "Poly MvE" (left), "Poly LvE" (middle) or "Poly LvM" (right) comparison in each cell type corresponding to either mid- (left) or late-phase snRNAseq clusters (middle and right, respectively) after using minimum mean expression by cluster (> 0.1) with at least 30% of cells in a cluster expressing the gene. Log2FC depicted using a color scale with |log2FC | > 1 and adjusted *p* value < 0.05. **g** GOCircle plots show PANTHER enrichment analyses using translationally changed mRNAs in (left) the early "SP&ExN L5/6" (E0, E7, E9; dark gray) and "ExN L5/6" (E5; light gray) clusters, and (right) the late "SP" (L4, L15, L16; dark gray) and "ExN L5/6" (L7, L10, L13, L14, L18; light gray) clusters. The gene list was obtained from "Poly LvE/early or late snRNAseq clusters" intersection analysis presented in Supplementary Data 5. The inner circle is a bar chart where the height of the bar represents the significance of the GO term (FDR < 0.05), color-coded by the z-score (the number of derepressed mRNAs minus the number of repressed mRNAs divided by the square root of the total number of counts). The outer circle denotes the log2FC scatter plot for each assigned derepressed (red dot) and repressed (blue dot) mRNA in each term. Tables below list information (ID, term description, fold enrichment (F.E.) value) for all significant specific subclass GO-CC terms, ranked by FDR (<0.05).

risk for NDDs, we conducted risk-gene set enrichment analyses using permutation-based statistics across all clusters leveraging ASD risk genes emerging from exome sequencing analyses[43] and genes associated with developmental delays curated by the Development Disorder Genotype - Phenotype Database (DDG2P)[54] (Fig. 1b, c). For the DDG2P, we restricted to genes with brain involvement (see Methods). In agreement with previous reports[43,44], we found an enrichment of ASD risk genes[43] in maturing and mature excitatory and inhibitory

neurons, as well as in progenitors (Fig. 1b-c). Importantly, we found a striking enrichment of ASD risk genes (e.g., *SCN2A* and *GRIN2B*[37,44]), but not DDG2P genes in SP clusters at both early fetal and midfetal periods (Fig. 1b-c; Supplementary Data 2). The high expression of ASD genes in the SP is also corroborated by a published mouse dataset[52]. These data support previous evidence that the midfetal cortical development is a nexus of high vulnerability for ASD and NDDs[43,44,47–51] and further implicate SP neurons in the etiology of ASD.

## Dissecting the developmental translational dynamics in human fetal cell types

Transcription largely shapes cell type identity during prenatal neocortical development (Fig. 1a) but gene expression is also greatly affected by post-transcriptional regulatory mechanisms, including mRNA translation. In fact, our previous work showed the important role of mRNA translation in cortical development[25,26,55–58]. To investigate translational changes that occur during human corticogenesis, we performed polysome profiling of human neocortices across early fetal (11/12 PCW), early midfetal (14/15 PCW) and late midfetal (17/18/20 PCW) periods. We then conducted standard RNA sequencing to obtain information on the distribution of mRNAs along the polysome profile (GEO accession GSE214272). Standard RNA sequencing may not be the most suitable approach for studying translational changes that could be influenced by alterations in the cell-type composition during neocortical development; however, it remains a valuable and commonly used tool that can overcome the technical limitations of polysome fractionation, which requires cell lysis in a polysome-extraction buffer. The polysome profiles showed separation of monosome-pooled fractions (40S-60S-80S) and polysome-associated (putatively actively translating) mRNAs from all gradient fractions (Supplementary Fig. 2a). Comparing the levels of free mRNAs (input) with those bound to polysomes revealed that both transcriptional and translational control of gene expression are dynamically regulated during the development of the human neocortex (Fig. 1d, Supplementary Data 3).

We identified a set of mRNAs under translational control across developmental phases that have comparable levels of expression in the input but show altered association with monosomes or actively translating polysomes depending on the phase (Fig. 1d, Supplementary Data 3). For example, the mRNA encoding the microcephaly-associated protein ASPM had significantly lower association with polysomes in late midfetal compared to early midfetal phase (Fig. 1d). We define less translationally active mRNAs as "translationally repressed" because of their higher association with monosome than polysomes, while "translationally derepressed" mRNAs indicate mRNAs that have higher association with polysomes than monosomes. Of the 13,586 transcripts with stable expression in input, 1648 mRNAs were differentially changed when comparing early fetal to early midfetal phases (825 derepressed and 823 repressed), and 745 mRNAs changed from early midfetal to late midfetal periods (374 derepressed and 371 repressed) (Fig. 1e and Supplementary Data 3). These results imply that the early-to-mid transition undergoes more dynamic changes in translation than the mid-to-late transition, as evidenced by higher number of mRNAs whose distribution shifted from monosomes to polysomes (increased translational efficiency) or vice-versa (decreased translational efficiency). Gene ontology (GO) enrichment analysis with clusterProfiler package[59] also showed that over-represented terms in both derepressed and repressed groups differed between early-to-mid and mid-to-late transitions (Supplementary Fig. 2b, Supplementary Data 4). These data suggest that highly dynamic mRNA translation mechanisms contribute to human neocortical development.

To get a snapshot of the complex translational status of each cell type across human developmental periods, we compared our polysome profiling datasets with our snRNAseq screens. Therefore, we considered mRNAs with $|\log_2 FC| > 1$ in the polysome profiling data, >0.1 minimum mean expression by snRNAseq cluster threshold and at least 30% of cells in a cluster expressing the mRNA. This direct comparative analysis was based on the assumption that single nucleus transcriptome profiles recapitulate faithfully many of the transcriptomic changes found in intact single cells[60,61], as carefully demonstrated by other side-by-side studies showing high correlation between single nucleus and whole cell transcriptomes[62–64]. By intersecting the snRNAseq datasets with the transcripts showing changes in monosome association, we identified 83 derepressed and 9 repressed

mRNAs in early-to-mid transition, and 327 derepressed and 33 repressed mRNAs in early-to-late transition (Supplementary Fig. 3, left and middle, respectively). However, mid-to-late monosome comparison showed small level of translational changes with only 2 derepressed and 5 repressed mRNAs (Supplementary Fig. 3, right). Focusing on transcripts with changes in polysome association, we identified 73 derepressed and 16 repressed mRNAs in early-to-mid transition, 109 derepressed and 58 repressed mRNAs in early-to-late transition, and 3 derepressed and 14 repressed mRNAs in mid-to-late comparison. The mRNAs bound to different populations of ribosomes (monosomes and polysomes) in each snRNAseq cluster across the three developmental phases are listed in Supplementary Data 5.

In summary, these profiling results suggest that spatiotemporal control of neocortical translation is high during early-to-mid transition, and then declines from mid-to-late phases. Some translationally regulated transcripts display cell type-restricted expression (i.e. *CACNG8, GPC5, KCND3, NUCB2* in monosomes; *ZNF385B, CDKL2, ZFR2, COTL1* in polysomes). Most, though, lack cell specificity (i.e. *ANKIB1, KMT2E, LSAMP, NETO2, OGA* in monosomes; *LMO3, GNG2, LIMCH1, UCHL1* in polysomes). The latter transcripts with ubiquitous expression may be crucial for facilitating or maintaining the cellular functions in all cell types in the neocortex. Overall, this analysis allowed us to assign translation-regulated transcripts to a specific human neuronal cell type during development, laying the foundation for future ad-hoc investigations.

Although monosomes are recognized as significant sites of active translation[65,66], our study focused on polysome-occupying mRNAs that synthesize multiple copies of a new protein from a single mRNA molecule[67,68], making them a canonical source of translation for dynamically evolving neocortical development[25,26,56]. To determine the "Cellular Component" (CC) terms associated with both derepressed and repressed differentially expressed genes (DEGs) that change from early to late polysomes, we performed the GO enrichment analysis on early and late excitatory neuronal clusters (Supplementary Data 5) using PANTHER Classification System version 17[69]. While DEGs from early and late upper layer neuronal clusters did not yield any significant CC terms annotations, DEGs from SP and deep layer clusters showed substantial association with synaptic terms in both developmental groups. Specifically, we observed a significant enrichment of DEGs from the early "SP&ExN L5/6" clusters in two specific subclass terms highlighting "transmembrane reporter complex" and "synapse" (Fig. 1g, left, dark gray). Gene enrichments further progressed to "ionotropic glutamate receptor complex", "integral component of postsynaptic density membrane", "cation channel complex" and "glutamatergic synapses" in late "SP" clusters (Fig. 1g, right, dark gray). This suggests that fully developed SP tends to have greater expression of genes implicated in synaptic function. While DEGs from the early deep layer 5/6 neurons were primarily involved in "postsynaptic density membrane", "glutamatergic synapses" and "transmembrane reporter complex" (Fig. 1g, left, light gray), the late excitatory 5/6 clusters were enriched for "mitochondrial respiratory chain complex I" and "integral component of postsynaptic density membrane", with the latter showing higher fold enrichment compared to the early phase (Fig. 1g, right, light gray).

The enriched GO-CC terms associated with synapses in both derepressed and repressed mRNAs (Fig. 1g) aligns with the notion that synapses first form in the SP and subsequently appear in ascending fashion in layers 4–6 in developing mammalian neocortex[4,6,7,9,17]. To further validate these results, we utilized the synapse biology SynGO database[70] (https://syngoportal.org) and confirmed that translationally-regulated mRNAs in both early and late SP and deep layer clusters (from Fig. 1g) are enriched for synaptic terms. SynGO revealed that 17 (out of 111) and 13 (out of 73) genes from the early "SP&ExN L5/6" and "ExN L5/6" clusters, respectively, were uniquely mapped to SynGO annotated genes, as compared to "brain" selected

background (Supplementary Data 6). Similar analyses were performed with DEGs from late "SP" and "ExN L5/6" clusters, where 22 out of 129 and 20 out of 125 genes were mapped to SynGO synaptic proteins, respectively (Supplementary Data 6). Taken together, these data indicate that translational control mechanisms exist prenatally during human neocortical development. In fact, translational regulation supports the more rapid maturation of human SP compared to upper and lower neocortical layers, potentially by redefining its synaptic protein repertoire from the first to the second trimester of gestation.

## CELF4 is expressed in SP and excitatory layer 5/6 neurons

The SP neurons are one of the first to be born and earliest to mature neurons in the developing neocortex[1,4,6,7,9,11,12,17,52], as further confirmed by our cell type identification in snRNAseq analyses. Their highly advanced maturity status, which can partially be attributed to the abundant expression of genes related to synaptic functions, lays the foundation for life-long cortical circuits. Seeking to identify genes that are expressed in both SP and layer 5/6 neurons, and can potentially regulate the translation of synaptic genes, we narrowed the scope of our investigation to RBPs, which are the major regulators of mRNA translation[18–21]. To identify candidate RBPs enriched in SP and associated with predicted synaptic function, we conducted DEG analyses between SP clusters and other neuronal clusters (Fig. 2a), separately for each developmental phase. We then filtered the identified DEGs by mapping them to GO terms associated with synaptic function and counted the number of terms associated with each gene. Moreover, we compared the DEG lists for each developmental phase with a collection of 6,100 human RNA-binding protein (RBP) candidates from the RBP2GO database[71] to determine which RBPs were significantly upregulated in the SP clusters. After ranking the RBPs by their RBP2GO scores, we found that CUGBP Elav-Like Family Member 4 (CELF4) was the top-ranked RBP in all three developmental phases (Supplementary Data 7). Importantly, some *CELF4*-expressing clusters (Supplementary Data 1) were also found to be enriched in the expression of ASD risk genes (Fig. 1c), further supporting the evidence of CELF4 as a ASD risk gene[72].

Next, we wanted to determine if *CELF4* expression in the SP and layers 5/6 neurons (Fig. 2a) models CELF4 protein expression in human developing neocortices between 10 and 21 PCW (Fig. 2b and Supplementary Fig. 4a). We labeled layer-specific cell types using established marker BCL11B/ CTIP2 (predominantly layer 5 and SP neurons; Fig. 2b) and TLE4 (predominantly layer 6 and SP neurons; Supplementary Fig. 4a). At 10 PCW, CELF4 positive (+) cells were strongly expressed in the MZ, while their distribution was unevenly spaced in the CP with more dominant expression in the presubplate and deep layers – where CELF4+ neurons clearly colocalized with CTIP2 nuclear staining (Fig. 2b; inset). During early mid-gestation (15 PCW), the SP exhibits sublaminar organization, which stems from its dramatic expansion compared to the CP. The CELF4+ cells were thus most apparent in the superficial subplate (SPs) where they colocalized with CTIP2 and TLE4 (Fig. 2b and Supplementary Fig. 4a; arrowheads in insets). Less intense CELF4 protein signal was detected in the deep subplate (SPd) and deep layers. By 17 PCW, the SP continues to delineate and increase in thickness. At that phase, CELF4+ neurons were primarily located in SPs (arrowheads in insets) and deep cortical layers 5/6, as evidenced by their colabeling with both CTIP2 and TLE4 (Fig. 2b and Supplementary Fig. 4a). At 21 PCW, the strongest expression of CELF4+ neurons was detected in the deep cortical layers 5/6; however, sparse but strong expression of CELF4, showing a prominent colocalization with CTIP2 and TLE4, was also noticeable in the fully expanded SP with gradual dilution of the signal across SPs (arrowheads in insets), intermediate subplate (SPi) and SPd (Fig. 2b and Supplementary Fig. 4a). Additionally, the CELF4 protein expression coincides with the expression of established SP enriched markers – NR4A2, ST18 and SERPINI1[22,73]. At 17 PCW, some cells in the developing SP showed co-expression between CELF4 and NR4A2 proteins (Fig. 2d, left; inset). At 18 PCW and 19 PCW, CELF4+ neurons in the SP strongly colocalized with *ST18* mRNA (Fig. 2c), while overlapping expression between CELF4 protein and *NR4A2* mRNA was less prominent in the SP neurons (Fig. 2d, right). Lastly, CELF4 immunoreactivity was also detected in the SERPINI1+ migrating and maturing SP neurons at 15 PCW, 17 PCW and 21 PCW (Supplementary Fig. 4b; insets). In addition to validating the relevance of our results for human neocortical development, these findings suggest that CELF4 can serve as novel and reliable marker of human SP and deep layers throughout neocortical development.

## Synapse-associated transcripts are bound by CELF4 in human fetal neocortex

To identify human mRNA targets of RBP CELF4 during neocortical development, we conducted native RNA immunoprecipitation (RIP) on early fetal (11/12 PCW), early midfetal (14/15 PCW), and late midfetal (17/18/20 PCW) neocortices using the validated CELF4 antibody or corresponding negative control immunoglobulin G (IgG). RIP samples were then subjected to RNAseq (GEO accession GSE214327). Using stringent criteria [$\log_2$FC (CELF4/IgG RIP) > 1.7 and adjusted $p$ value < 0.05], we observed a total of 227 target mRNAs in the early phase, while a much higher number of putative CELF4 targets was detected in the mid (a total of 1536 mRNAs) and late (a total of 763 mRNAs) phases (Fig. 2e and Supplementary Data 8). Among these mRNAs, 45 early, 183 mid and 125 late CELF4 candidates encode presynaptic and postsynaptic proteins, based on SynGO annotations[70] (Supplementary Data 8). By using phase-specific CELF4 targets, the significant theme from the top eight GO-BP PANTHER[69] results was enriched for subclass terms related to synapse, neurotransmission, cell projection morphogenesis and transport, as depicted in chord diagrams for each developmental phase separately (Fig. 2f). Furthermore, 175 candidate mRNAs was shared between the three developmental points, with 36 mRNAs showing highly significant association with synapse functions based on the SynGO database (32 mRNAs were associated with GO-CC descriptor "synapse", 21 mRNAs with the term "presynapse" and 7 mRNAs with "synaptic vesicle membrane"; Supplementary Data 8).

To validate some of these putative CELF4 mRNA targets, we performed quantitative real-time PCR (qRT-PCR) using RNAs of RIP-RNAseq experiments. Sequence-specific primers (Supplementary Data 11) were designed for several CELF4 target candidates encoding presynaptic (*SYNPR, SYN2, SV2A, SYP, APBA1*) and postsynaptic (*GABRA3*) proteins, transcription factors (*TLE4, NR4A2, ST18*), or translation initiation factors (*EIF4A2*) in the SP area. Enrichment of selected mRNA targets by RIP-qRT was calculated as the fold-change bound by CELF4 from the non-specific binding in IgG RIP. Since the remaining quantities of purified RNAs from each developmental-phase specific CELF4 RIPs were limited after RNAseq, we grouped together the results of all developmental phases obtained from each RIP-qRT experiment. As expected, the expression levels of internal control *ACTB* and negative control *NES* were not found enriched in CELF4 RIP samples, confirming the specificity of the approach. In contrast, selected CELF4 mRNA candidates (*SYNPR, SYN2, SV2A, SYP, APBA1, GABRA3, TLE4, NR4A2, ST18* and *EIF4A2*) were significantly enriched in CELF4 RIPs (Fig. 2g). Some of these mRNAs were monitored for expression in the developing fetal neocortex and showed co-localization with CELF4 protein in the SP area (Supplementary Fig. 4c). These results confirm that CELF4 binds a set of synapse-associated mRNA targets in vivo in human developing neocortices.

Among validated CELF4-targets is the mRNA encoding the synaptic vesicle glycoprotein 2 A (SV2A). SV2A is highly expressed in all glutamatergic and GABAergic cortical subtypes, both of which are also present in the developing SP[4–7,9–14,17]. Recent studies have identified presynaptic protein SV2A as a promising biomarker for the quantification of synaptic density in various neurological conditions,

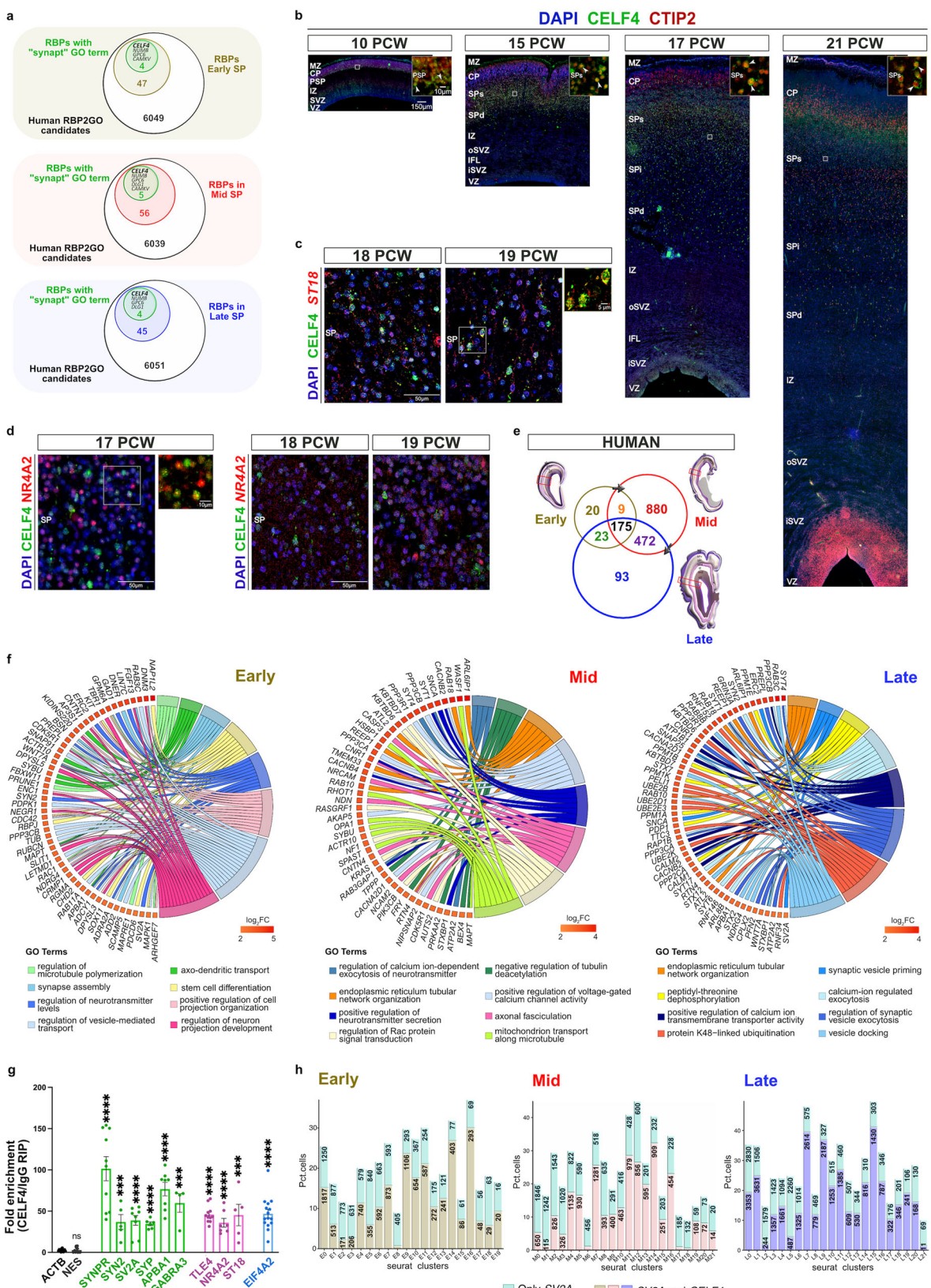

underscoring its importance in normal brain functioning[74,75]. Therefore, we sought to examine the percentage of nuclei expressing *SV2A* mRNA alone or in combination with *CELF4* from the total number of nuclei in each cluster across early, mid and late phases of neocortical development (Fig. 2h). Although *SV2A* mRNA was detected in all nuclei clusters, the percentage of nuclei expressing *SV2A* varied across

developmental phases. Notably, excitatory neuronal clusters consistently displayed a high number of *SV2A*-expressing cells, while the non-neuronal clusters displayed the opposite trend. Out of total number of nuclei with detectable *SV2A* expression, the SP and excitatory layer 5/6 clusters showed relatively high co-expression of *SV2A* and *CELF4*, ranging from 81% to 59% in the early phase, 75% to 53% in

**Fig. 2 | CELF4 is expressed in deep cortical plate and synapse-rich subplate during human neocortical development. a** Venn diagrams show upregulated RBPs in subplate (SP) clusters versus all other neuronal clusters after pre-filtering Supplementary Data 7 for DEGs that (1) are present in the RBP2GO list, (2) have avg_log2FC > 0 (up-regulated), (3) have one or more GO-BP terms associated with synaptic function, (4) have RBP2GO scores > 0. CELF4 ranked highest in all developmental phases. **b** Representative images of CELF4 (green) and CTIP2 (red) protein expression in fetal prospective dorsolateral prefrontal cortex at 10, 15, 17 and 21 PCW. DAPI is in blue. Enlarged insets of the gray boxes in the main figures show colocalization of CELF4 immunoreactivity with CTIP2+ neurons (arrowheads) in the SPs. Scale bar of 10x and 40x (inset) objective lens: 150 μm and 10 μm, respectively. CP cortical plate, MZ marginal zone, PSP presubplate, SPs superficial subplate, SPd deep subplate, SPi intermediate subplate, IZ intermediate zone, oSVZ outer subventricular zone, IFL inner fibrillar layer, iSVZ inner subventricular zone, VZ ventricular zone. **c** ImmunoFISH confirms CELF4 protein (green) colocalization with *ST18* (red) and (**d**) *NR4A2* (red) mRNAs in the SP of developing human fetal cortices. Scale bar of 60x objective lens: 50 μm or 5 μm (inset). **d** Confocal images show CELF4 protein (green) and NR4A2 protein (red) colabeling (inset) in the SP of the 17 PCW developing neocortex. Scale bar of 20x and 80x (inset) objective lens: 50 μm and 10 μm, respectively. **e** Venn diagram illustrates intersection of human CELF4 (hCELF4) targets across key phases of neocortical development. Total of 227, 1536 and 763 hCELF4-targets were identified in the early (ochre; n = 3 neocortices), mid (red; n = 2 neocortices) and late (blue; n = 3 neocortices) phases using log2FC > 1.7 with adjusted p value < 0.05. **f** GOChord show the most enriched PANTHER GO-BP for hCELF4 targets identified in early (left), mid (middle) and late (right) phases from (**e**). Target mRNAs are ordered according to their log2FC value from highest (red) to lowest (yellow). **g** Bar graph represents fold enrichment of hCELF4 targets from early (n = 3 neocortices), mid (n = 2 neocortices) and late (n = 3 neocortices) hCELF4 RIPs using specifically designed qRT-PCR primers (Supplementary Data 10) for synapse-associated genes (*SYNPR, SYN2, SV2A, SYP, APBA1, GABRA3*), transcriptional factors (*TLE4, NR4A2, ST18*) and translation initiation factor (*EIF4A2*). *ACTB* is internal control. *NES* is negative (non-binding) control. Relative mRNA levels were normalized to the same ratio in the IgG RIPs. Data points, presented as mean and SEM, for each tested target were obtained after combining results from all developmental phases together (n = 4–8 cortices). qRT-PCR run with ≥2 technical replicates per each target and developmental phase. P values are shown in Source data. Statistical significance was determined by non-parametric unpaired, two-tailed Mann–Whitney test. ***$p \leq 0.001$ and ****$p \leq 0.0001$. **h** Stacked bar plot illustrates the percentage of nuclei expressing *SV2A* alone (turquoise) or in combination with *CELF4* (tan, rose, slate blue), expressed as percent of all nuclei in each cluster across early, mid, and late phases of cortical development. The actual nuclei count values are labeled within each bar for each cluster.

the mid phase and 83% to 60% in the late phase (Fig. 2h). Given that both transcripts are frequently co-expressed in the same cell, it is plausible that the role of RBP CELF4 and SV2A could converge on synaptic functions.

## Celf4 is expressed in the subplate during mouse embryonic cortical neurogenesis

Given the specificity of CELF4 protein expression in the SP and layer 5/6 neurons in human developing neocortices (Fig. 2b), we investigated whether this protein expression pattern remains conserved during early [embryonic day 11.5 (E11.5), E13.5] and later stages [E15.5, E17.5, postnatal day 0 (P0)] of mouse corticogenesis. To obtain a more precise understanding of Celf4 distribution in relation to specific neuronal identities, we analyzed the co-localization of Celf4 with Reelin (Cajal-Retzius marker), Nr4a2/Nurr1 (the SP marker), and deep layer markers Bcl11b/Ctip2 (mostly layer 5) and Foxp2 (mostly layer 6) by double immunohistochemistry (IHC). At E11.5, when the peak neurogenesis of Cajal-Retzius cells occurs, Celf4 expression was noticeable in the MZ as revealed by its colocalization with Reelin protein. The significant overlap of Celf4+ and Reelin+ neurons remained visible in the MZ throughout the subsequent stages of neocortical development (Supplementary Fig. 5a). In contrast, only a few weakly-labeled Celf4+ neurons were observed in the preplate at E11.5 (the production peak for SP neurons) and in the SP at E13.5 (the onset of preplate splitting), some of which also co-expressed Nr4a2 and/or Ctip2 markers (Fig. 3a). In contrast, prominent expression of Celf4 by SP neurons became obvious from E15.5 onwards. At this age, SP neurons that co-expressed both Celf4 and Nr4a2 only expressed the latter at lower levels. While Celf4 protein expression also occurred in the population of Ctip2+ neurons in the SP at E15.5, very rarely Celf4+ neurons were found within the CP. At E17.5, faint Celf4-expressing cells became dispersed only in the deep layers of the CP, but strongly labeled Celf4+ cells were increasingly restricted to SP region and clearly co-localized with both Nr4a2 and Ctip2. At both E15.5 and E17.5, almost all of the Celf4+ cells were localized in the SP area (Supplementary fig. 5b, left and middle) as indicated by the near-complete co-localization of Celf4 and Ctip2 staining (Supplementary Fig. 5b, right). At postnatal day 0 (P0), Celf4 protein expression remained mostly confined to developing SP, where it co-localized with the Nr4a2 that is organized as a narrow band in the SP. At this stage, the number of weakly expressing Celf4+ cells were also increased in the deep cortical layers. Here, Celf4 expression co-localized with Ctip2+ cells that mostly reside in layer 5 (Fig. 3a) and Foxp2+ neurons that are predominantly localized in the layer 6 (Supplementary Fig. 5c). These findings suggest that the distribution pattern of Celf4 expression is evolutionarily conserved between human and mouse developing neocortices.

Since SP neurons make the first neocortical synapses laying the foundation for mature circuits, we examined the Celf4 expression in neurons that carry synapses in the developing mouse and human SP (Fig. 3b). Using Celf4 immuno-electron microscopy, we detected early synaptic contacts on Celf4+ SP neurons in E15.5 and E17.5 mouse SP, as well as on CELF4+ SP neurons in 19 PCW human SP zone. These results suggest that Celf4 has an important and potentially conserved function at the level of SP synapses during both mouse and human neocortical development.

## Intersection of human and mouse Celf4 RIP-RNAseq findings from developing neocortices

To determine if there is a conservation of target mRNAs that are bound by human CELF4 and murine Celf4, we identified and characterized common mRNA candidates after intersecting our human RIP-RNA seq data sets with our mouse RIP-RNA seq data (Supplementary Data 9). In the latter, we employed previously described RIP-RNAseq strategy on mouse E17 neocortices (n = 3, log2FC > 0.5 and adjusted p value < 0.05). This developmental time point was selected because of strongly confined neocortical expression of Celf4 in the SP. Additionally, all layer 5/6 neurons have been generated and have acquired their laminar position at E17. Furthermore, human and mouse brain maturation can be paralleled using various criteria, such as cortical neurogenesis, neuronal migration, or synaptogenesis[76-78]. For this reason, we performed three comparisons where mouse Celf4 targets were intersected with human CELF4 targets identified in early, mid or late developmental phases, yielding a total of 102, 220 and 201 common target mRNAs, respectively (Fig. 3c).

To investigate the functional role of Celf4/CELF4 in both mouse and human developing neocortices, we analyzed for overrepresented GO categories from the shared targets using PANTHER[69] (Fig. 3c). The top nine GO-BP terms in all three comparisons revealed an enrichment for synaptic functions (e.g., "positive regulation of filopodium assembly", "synapse assembly", "modification of synaptic structure", "neurotransmitter secretion", "synaptic vesicle localization" and "synaptic vesicle transport"), suggesting a highly conserved role of Celf4/CELF4 in the formation and maintenance of neocortical synapses. Furthermore, we verified a subset of putative shared targets by qRT-PCR on a separate mouse Celf4-immunoprecipitates (Fig. 3d). The mRNA levels of negative controls, *Actb* and *Nes*, were not found to be enriched in the RIPs compared to the internal control *Gapdh* [log2FC (Celf4/IgG RIP)]. Unlike *Adra2a,c* mRNAs that served as synapse-specific negative

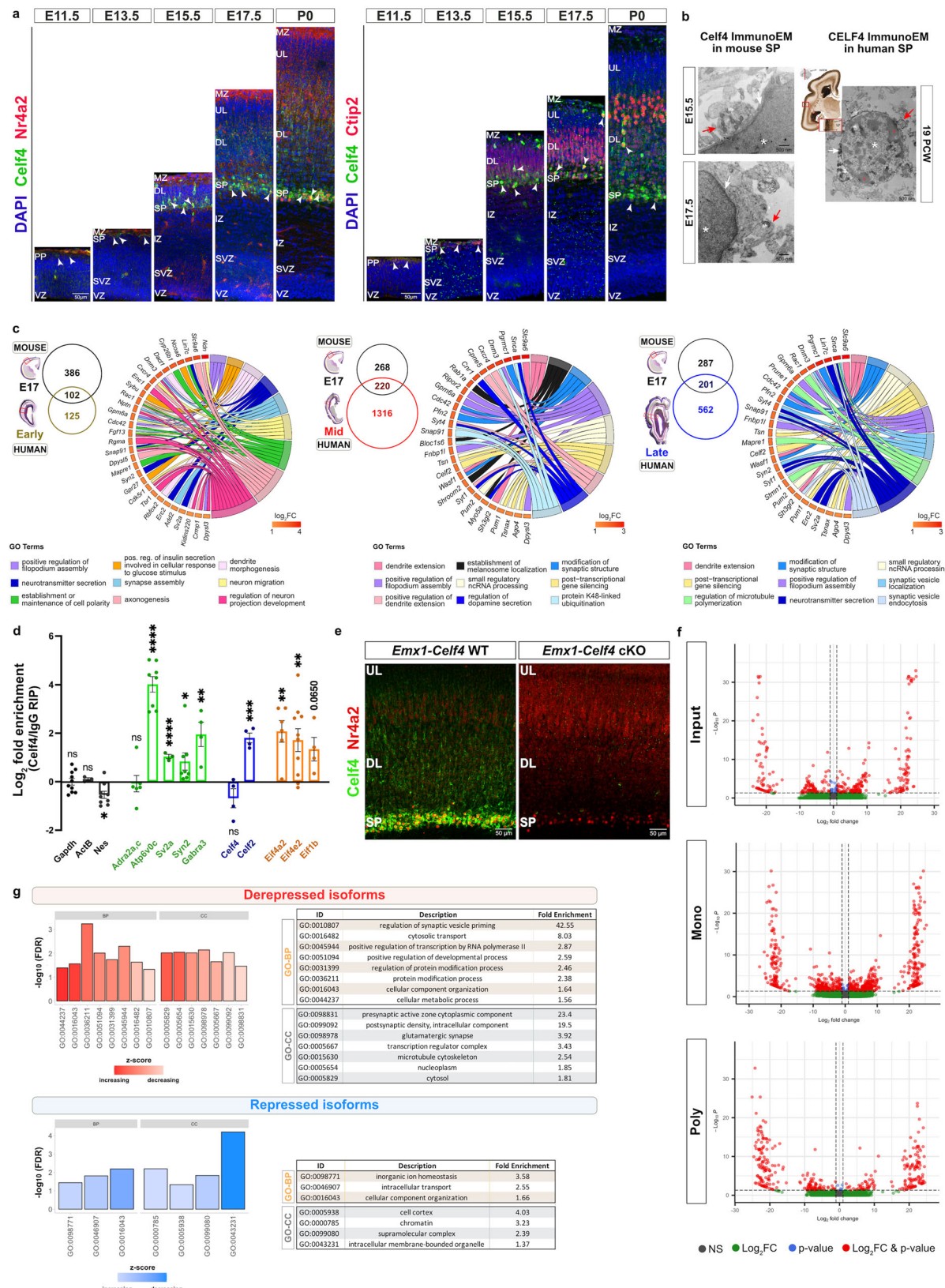

control, each of the shared synapse-associated mRNAs (*Atp6v0c, Sv2a, Syn2, Gabra3*) showed significant enrichment in the Celf4 RIPs (Fig. 3d), confirming a role of Celf4 in the regulation of the key genes for synaptic development in the neocortex. *Celf4* mRNA did not enrich its own mRNA. Another member of the same family, Celf2, was identified as a common target from our intersection analysis (Fig. 3c), and

was validated by qRT-PCR (Fig. 3d), indicating a potential cross-regulation between RNA-binding proteins. We also validated mRNA targets encoding translation initiation factors *Eif4a2* (identified as common target), *Eif4e2* (initially identified as human CELF4 target), and *Eif1b* (identified as mouse-specific target) (Fig. 3d). Overall, these data corroborate the validity of the mRNAs identified by RIP-RNAseq

**Fig. 3 | Expression and function of Celf4 during mouse neocortical development. a** Confocal images of mouse neocortices at E11.5, E13.5, E15.5, E17.5 and P0 showing Celf4 (green) co-localization (arrowheads) with Nr4a2 (red, left) and Ctip2 (red, right) in the SP and layers 5/6. DAPI shown in blue. Sample size: n = 3 animals per developmental stage. Scale bar of ×20 objective lens: 50 μm. **b** Immuno-EM showing early neocortical synaptic contacts on E15 (n = 3 animals) and E17 (n = 2 animals) mouse Celf4-positive SP neurons (asterisk, left) and human CELF4-positive SP neurons at 19 PCW (asterisk, right). The Celf4/CELF-immuno-labeled SP neuropil is surrounded with a large extracellular matrix space. Middle images from Bayer&Altman[101] show the cutting spot in the 19 PCW brain (red line) and the area where the SP was dissected from the temporal cortex, since frontal cortex tissue was unavailable for EM (knife). Red arrows point to prospective pre-synaptic element. White arrows mark possible synaptic contacts. Red circles denote clumps of chromatin on the periphery of the cell nucleus. Mt denote mitochondria. Scale bar: 500 nm (magnification for mouse EM micrograph: 8000x; for human EM micrograph: 5000x). **c** Venn diagrams display the intersection of 488 mouse Celf4 (mCelf4) targets (log$_2$FC > 0.5 and adjusted p value < 0.05) with early- (left), mid- (middle) and late- (right) phase hCELF4 targets (from Fig. 2e). Sample size: n = 3 separate mCelf4 RIP-RNAseq (6–9 E17 neocortices processed as single biological sample). GOChords depict the most enriched specific PANTHER GO-BP terms for each gene list: "mCelf4/early hCELF4" (left), "mCelf4/mid hCELF4" (middle) and "mCelf4/late hCELF4" (right). mCelf4/hCELF4 targets are ordered according to their log$_2$FC value from highest (red) to lowest (yellow). **d** qRT-PCR validation of shared mCelf4/hCELF4 targets using the leftover RNA from mCelf4 RIP-RNAseq. These targets are associated with synapses (*Atp6v0c, Sv2a, Syn2*, and *Gabra3*), are

RNA-binding proteins (*Celf2*), or function in translation initiation (*Eif4a2, Eif4e2, Eif1b*). Celf4 did not enrich its own mRNA, nor *Nes* mRNA, serving as a negative (non-binding) control. *ActB* was used as an internal control. *Adra2a,c* mRNAs were used as synapse-specific negative control. Sample size: n = 3–5 animals. qRT-PCR run with ≥3 technical replicates per each target. Data represent the mean and SEM. P values are shown in Source data. Statistics: unpaired, two-tailed Welch's t test for parametric analysis, or Mann-Whitney test for non-parametric analysis. ns p > 0.05, *p ≤ 0.05, **p ≤ 0.01, ***p ≤ 0.001, ****p ≤ 0.0001. **e** Immunostaining with Celf4 (green) and Nr4a2 (red) antibodies to confirm successful generation of *Emx1-Celf4* knockout (cKO) line. Sample size: n = 3–4 animals. UL upper cortical layers. Scale bar of 60x objective lens: 50 μm. **f** Volcano plot of differentially expressed isoforms in P0 *Emx1-Cre Celf4* cKO relative to WT (n = 3 spins per condition, 3 neocortices pooled together as one biological sample) in various fractions: input (top), monosome fraction (middle) and polysome fraction (bottom). Red circles, isoforms that are significantly changed by |log$_2$FC| > 1 and adjusted p value < 0.05 in the input, monosome or polysome fractions. Gray dots, unchanged isoforms (threshold marked as horizontal dashed line). P values shown in the plot were estimated by a two-tailed Wald test in DESeq2 and adjusted for multiple comparisons using the procedure of Benjamini-Hochberg. **g** GO bar plots show the most enriched GO-BP and GO-CC subclass terms, sorted by the z-score and represented as -log$_{10}$ of FDR for derepressed (top) and repressed (bottom) isoforms in the polysome fractions (from **f**, bottom). Tables display information (ID, term description, fold enrichment) about the PANTHER significant subclass GO-BP (orange) and GO-CC (gray) terms.

are bona fide targets of Celf4/CELF4 in human and murine developing neocortices.

## The function of Celf4 during mouse neocortical development

We next wanted to assess the birth date of neurons expressing Celf4. With this aim, we performed sequential labeling of dividing apical progenitors with chloro-deoxyuridine (CldU) at the onset of neurogenesis (E10.5) and iodo-deoxyuridine (IdU) at the start of SP neurogenesis (E11.5). Analysis at E17.5 revealed that only a few Celf4+ neurons were born at E10.5, and more of them were born at E11.5. Both of E10.5- and E11.5-born Celf4+ neurons were distributed predominantly in the developing SP, and few migrated into the MZ (Supplementary Fig. 6a). We then performed the double labeling experiment at E12.5 (CldU) and E13.5 (IdU) to capture the laminar positions of Celf4+ neurons at the time of preplate splitting. While most Celf4+ neurons born at E12.5 were destined for SP, some started to populate deep cortical layers. In contrast, Celf4+ neurons born at E13.5 were mostly absent from SP and were primarily detected in the deep cortical layers (Supplementary Fig. 6b). These data suggest that Celf4+ neurons are among the earliest born and maturing neurons in the developing neocortex.

To start addressing the neurodevelopmental role of Celf4 and its potential contribution to the functional synaptic circuits in the SP, we selectively deleted *Celf4* (B6.129-Celf4$^{tm1.1Frk/Frk}$) from cortical neural progenitors and their neuronal progeny starting from E9.5 by utilizing the *Emx1*-Cre;*Celf4*$^{fl/fl}$ (*Emx1-Celf4*) conditional knockout (cKO) line[79,80]. The loss of *Celf4* from the SP and cortical layers was confirmed at the protein level with IHC (Fig. 3e) and Western blot at P0 (Supplementary Fig. 7a). Since SP regulates the fate selection of deep layer neurons[81] and the multipolar-to-bipolar switch in the neuronal migration during the mid-embryonic phase[82,83], we first analyzed the neocortical architecture of *Emx1-Celf4* cKOs and their littermate controls at P0. This developmental stage was chosen because all neuronal subtypes are completely separable based on their laminar positions at P0. We performed double-IHC for Bcl11b/Ctip2 and Satb2 to label distinct subpopulations of deep layer neurons and intracortically projecting neurons, as well as double-IHC for upper layer neuronal subtypes using established markers Pou3f3/Brn1 and Cux1/CDP. The density and laminar placement of these distinct subtypes of neurons were not affected upon *Celf4* deletion (Supplementary Fig. 7b, c).

We therefore postulated that Celf4 operates far beyond the regulation of cell fate and laminar distribution in the SP and deep cortical layers, and its role may extend to other developmental events. Previous studies showed that the SP plays a pivotal role in coordination of thalamocortical synapse formation during early neocortical development[14,16,17]. Since Celf4 binds a set of mRNAs involved in synaptic functions (Fig. 3d) and shows a strong expression in the developing SP (Fig. 3e), we reasoned that Celf4 might influence the development of earliest neocortical synaptic contacts and/or modulate their synaptic properties. Thus, we first investigated whether Celf4+ SP neurons colocalize with pre- and post-synaptic markers of glutamatergic [vesicular glutamate transporter (vGlut2) and postsynaptic density protein 95 (Dlg4/PSD95)] and inhibitory synapses [vesicular GABA transporter (VGAT) and Gephyrin (Gphn)] starting at E15.5 when thalamic afferents begin to accumulate in the SP (Supplementary Fig. 8). At E15.5, Celf4+ neurons substantially co-localized in the MZ and the SP with both vGlut2 (a reliable marker of thalamic afferents), and PSD95 (an essential postsynaptic scaffolding protein in excitatory neurons). Likewise, Celf4/vGlut2/PSD95 co-labeling showed considerable overlap in the SP at E17.5, with significantly increased level of synaptic markers co-expression than what was observed at E15.5. While vGlut2 and PSD95 became more prominently expressed in the deep neocortical layers by P0, Celf4+ neurons remained strongly co-localized with these synaptic markers in the SP at P0 (Supplementary Fig. 8a). Similarly, co-localization of Celf4 protein expression with VGAT and Gphn staining was observed in the MZ and the SP at E15.5, which is indicative of the early development of GABAergic neurons during prenatal corticogenesis. At E17.5 and P0, the degree of Celf4/VGAT/Gphn co-labeling remained high in the SP, while a further population of Celf4+ neurons that also co-express VGAT and Gphn has emerged in the deep cortical layers by P0 (Supplementary Fig. 8b). These data show that Celf4 can be found in both glutamatergic and inhibitory neurons in the transient SP zone during the earliest stages of neocortical development.

## *Celf4* deficiency affects the polysomal positioning of synaptic target mRNAs

Given the role of RBPs in mRNA translation[18–21], the dynamics of mRNA translation during cortical development (Fig. 1), and the association of Celf4 with mRNAs critical for synaptic development (Figs. 2–3), we

conjectured that Celf4 regulates the translation of mRNAs important for developing neocortical synapses. To test this hypothesis, we performed unbiased polysome profiling-RNAseq on neocortices from P0 WT and *Emx1-Celf4* cKO neonates (Supplementary Fig. 9a; GEO accession GSE214328). Since Celf4 binds mRNAs for elongation initiation factors in an isoform-specific manner (Supplementary Fig. 9b), we implemented an isoform-level differential expression analysis of monosome (40S-60S-80S) and polysome fractions. The volcano plots generated from these data revealed two isoform groups, with one group showing changes in transcriptional stability, while the other group displayed stable transcriptional levels but had potential diversity in translational regulation (Fig. 3f). We were interested in the transcriptionally-stable isoform mRNAs in the latter group, as their association with monosomes and/or polysomes correlated with their expression levels. Specifically, we detected 729 isoform mRNAs with differences in association with monosome, 239 isoform mRNAs predominantly associated only with translating polysomes, and 89 isoforms with changes in association with both monosomes and polysomes. In the last two groups of isoform mRNAs with differential association with polysomes, 145 and 183 mRNA isoforms were translationally derepressed and repressed, respectively. (Supplementary Data 10). To further investigate the functional properties of these isoforms that are either enriched or depleted in polysome fractions, we used the GO enrichment analysis tool from PANTHER[69]. Interestingly, only translationally derepressed isoforms in *Emx1-Celf4* cKO were enriched for synaptic functions, as visible from the first enriched GO-BP ("regulation of synaptic vesicle priming") and GO-CC ("presynaptic active zone cytoplasmic component") terms from the total list of significant subclass GO terms (Fig. 3g). These data suggest that Celf4 might act as a translational repressor of synaptic mRNAs.

The mRNA expression changes from our polysome RNAseq screen were relatively modest at the gene-level, which has prompted us to calculate polysome/ monosome count ratio to serve as an approximate correlate for the translational activity within each polysomic group (Supplementary Data 10). To further confirm whether these transcripts identified in our RNAseq analysis are subjected to translational regulation, we calculated relative mRNA levels (polysome/monosome ratio) via qRT-PCR on previously sequenced sucrose gradient fractions. Several synaptic mRNAs (*Homer2, Sema4c, Ank2, Srcin1*), including Celf4 targets (*Map1b, Ppp1r9a, Sv2a, Syp, and Slc17a7/vGlut1*) had a polysome/monosome ratio significantly greater than $\log_2 FC > 0$, suggesting their distribution is shifted to polysomal fractions in *Emx1-Celf4* cKOs (Fig. 4a, Supplementary Fig. 9c). In contrast, some synapse-associated genes (*Dlg4/Psd95, Dmxl2, Gabra2*) were not translationally changed by Celf4, suggesting that Celf4-specific translational repression could play an important role during prenatal synaptogenesis. In contrast, the mRNA level of Celf4 target *Eif4a2* was strongly decreased in polysome/monosome ratio ($\log_2 FC < 0$), implying that *Eif4a2* mRNA is predominantly associated with monosomes when *Celf4* is knocked-out (Fig. 4a). Overall, these findings indicate that *Celf4* loss causes a modest but specific increase of translation for a specific subset of mRNAs critical for synaptic formation and function.

## Celf4 acts as a translational repressor of specific presynaptic transcripts

To further explore the neurodevelopmental role of Celf4 in translational regulation, we focused on translationally regulated Celf4 targets (translation initiation factor *Eif4a2* and pre-synaptic markers *Sv2a* and *Syp*) and tested whether they are expressed in the SP neurons and co-localize with Celf4 in both mouse and human developing neocortices. By combining immunohistochemistry and fluorescent in situ hybridization, we observed that Celf4 protein was not only strongly co-localized with *Eif4a2* mRNA, but also with its protein in the mouse SP neurons at P0 (Fig. 4b, top) and 19 PCW human SP (Fig. 4c). These

observations are consistent with our previous findings that Celf4 binds to *Eif4a2* mRNA (Figs. 2g, 3d) and translationally activates *Eif4a2* mRNAs (Fig. 4a). Since Celf4 and Sv2a primary antibodies have been raised in the same host species, we reasoned that we could use Eif4a2 protein as an additional marker to label the Celf4 expression in the developing SP due to its high expression level and co-localization with Celf4 protein in the SP.

Next, we observed that Celf4 protein co-localizes with pre-synaptic *Sv2a* mRNA and post-synaptic *PSD95* mRNA puncta (Fig. 4b, bottom), as well as pre-synaptic *Syp* mRNA puncta in the mouse neo-cortical SP zone at P0 (Supplementary Fig. 9a, left and right). Further, both human CELF4 and EIF4A2 proteins were co-localized with *SV2A* mRNA in the human SP zone at 19 PCW (Fig. 4d, left and right). In contrast, SV2A protein was almost always expressed in SP areas also expressing *SV2A* mRNA, but not EIF4A2 protein which was used to label Celf4 expression (Fig. 4d, middle and right). In the mouse SP at P0, pre-synaptic Sv2a protein expression was reduced in SP areas with the high expression of Eif4a2 proteins (used as marker of Celf4 protein expression; Fig. 4e, top). Similarly, Syp protein expression in the SP was very limited in the certain SP areas that show high co-localization of *Syp* mRNAs with Celf4 proteins (arrows in Supplementary Fig. 10a, middle and right). The *Syp* mRNA and its protein expressions were also stronger in the areas that display lower presence of Celf4 protein (arrowheads in Supplementary Fig. 10a). Analogously to SV2A protein expression in the human SP, the reduced expression pattern of human SYP protein was observed from the *SYP* mRNAs that were co-localized with CELF4 proteins in the SP neurons at 19 PCW and 21 PCW (Supplementary Fig. 10b). In contrast, the post-synaptic *PSD95* mRNA expression was detectable in the SP areas also expressing PSD95 proteins and CELF4 proteins (arrows in Supplementary Fig. 10c), supporting our polysome profiling data that *Dlg4/Psd95* mRNA is not translationally changed by Celf4 (Fig. 4a). These findings point out that RBP Celf4/CELF4 binds specific presynaptic mRNAs and represses their translation in the SP neurons of mouse and human developing neocortex.

We then set out to examine how the expression patterns of translation initiation factor Eif4a2, and presynaptic Sv2a and Syp proteins are changed in the SP zone of *Emx1-Celf4* mutants when compared to WT neocortices at P0. In line with our polysome association analysis (Fig. 4a), the Eif4a2 protein expression was evidently reduced while the Sv2a protein punctate staining was increased in the SP of *Emx1-Celf4* mutants (Fig. 4e, bottom). Interestingly, the selective deletion of *Celf4* lead to a decrease in the number of SP neurons showing both strong nuclear Eif4a2 and weak cytoplasmic Sv2a signal (Fig. 4e, middle). In contrast, the percentage of Eif4a2$_{weak}$+Sv2a$_{strong}$+SP neurons was increased in the mutants when compared to the littermate controls (Fig. 4e, right). Analogous to Sv2a protein changes upon *Celf4* deletion, the protein expression from *Syp* mRNAs was also increased in the SP of the *Celf4* cKO mouse neocortex at P0 (Supplementary Fig. 9d, bottom). Overall, our findings not only strengthen the suggestive link between ASD and both presynaptic markers Syp and Sv2a, but also suggest that Celf4 utilizes a spatially-specific translational regulation of its synaptic targets.

After confirming that *Sv2a* mRNA levels remained unchanged in total neocortical homogenates of *Emx1-Celf4* mutant at P0 (Supplementary Fig. 10e), we examined whether the subcellular levels of Sv2a protein are affected upon *Celf4* deletion. By using pan-Sv2 antibody, we performed Western blot analysis of Sv2/Sv2a protein in total neo-cortical homogenates and crude synaptoneurosomes isolated from WT and *Emx1-Celf4* cKO at P0. Although pan-Sv2 antibody recognizes all three isoforms (Sv2a, Sv2b, Sv2c), the Sv2a isoform is a predominant isoform in the rodent neocortex and specifically found in all neurons[84,85]. Our results showed that *Celf4* deletion did not alter Sv2/Sv2a protein levels in the total neocortical lysates (Fig. 4f, left) but the Sv2/Sv2a protein levels were significantly increased in the *Emx1-Celf4*

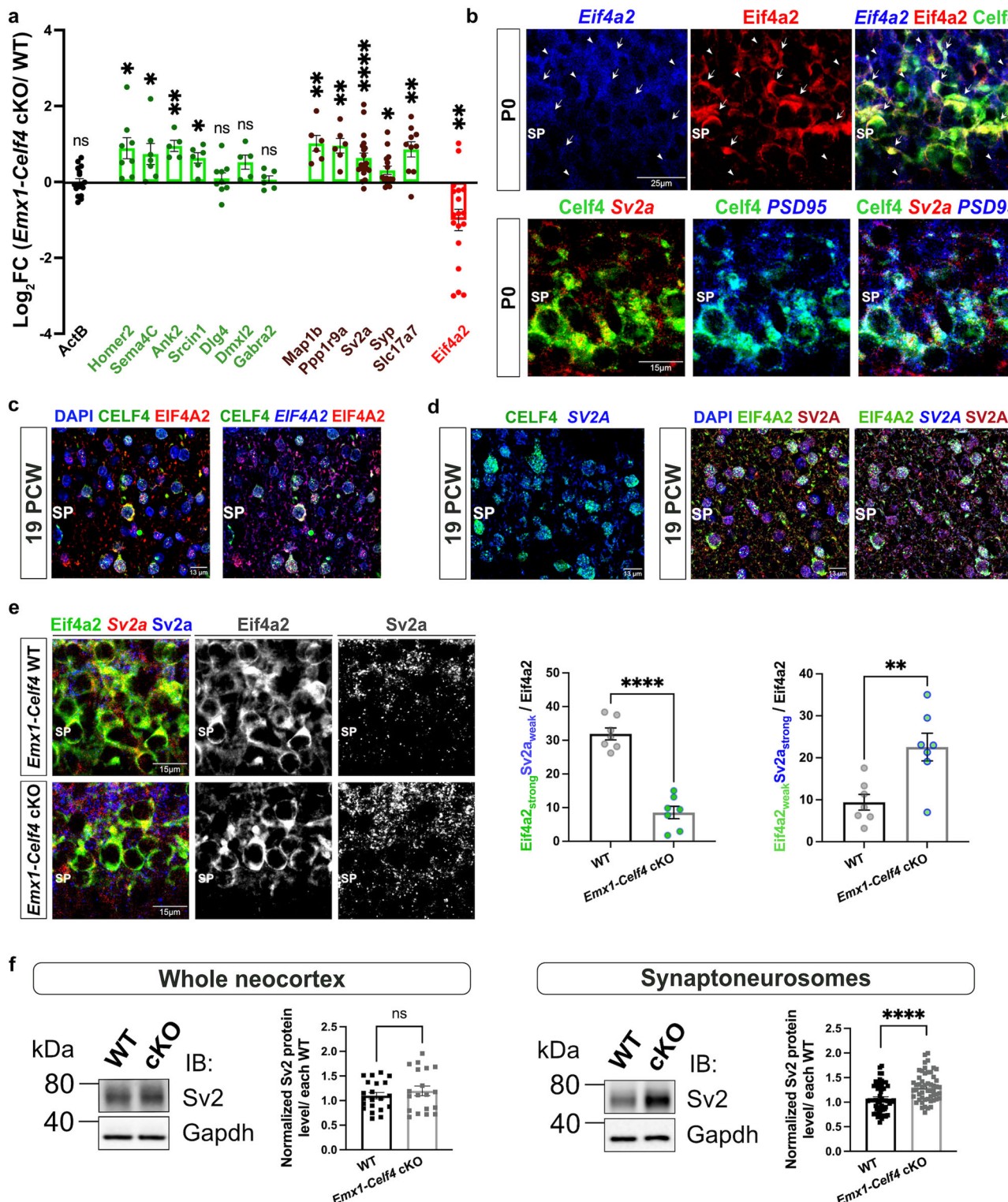

mutant synaptoneurosomes (Fig. 4f, right). Together, these data additionally show that Celf4-specific regulation of translation is found to account for significant differences in abundances of Sv2/Sv2a protein.

### Deletion of *Celf4* affects prenatal GABAergic and glutamatergic synapses in a sex-specific fashion

Since SP neurons provide a substantial GABAergic and glutamatergic synaptic input to the CP and subcortical areas[1,10–12,16,52,76], we decided to assess whether *Celf4* loss influences the prenatal formation of

GABAergic and glutamatergic synapses. To this end, we conducted unbiased and high-throughput analyses on P0 WT and *Emx1-Celf4* cKO neocortices using ThermoFisher CellInsight CX7. We examined GABAergic synapses (defined as VGAT+ puncta contacting Gphn+ puncta), and cortico-cortical or thalamo-cortical glutamatergic synapses (defined as vGlut1+ or vGlut2+ puncta, respectively, contacting PSD95+ puncta). Given the influence of sex on brain development[86] and ASD pathophysiology[87], we conducted analyses by sex. We found that non-nuclear VGAT+Gphn+ overlapping area was significantly increased in male *Emx1-Cre* Celf4 cKO neonates and

**Fig. 4 | Celf4/CELF4 binds to and translationally regulates synapse-associated targets in the subplate of mouse and human developing neocortices. a** qRT-PCR validation of translationally regulated mRNAs using polysomic RNAseq samples (*n* = 3 spins per condition, 3 pooled cortices as one biological sample). Synapse associated mRNAs (*Homer2*, *Sema4c*, *Ank2*, *Srcin1*), including Celf4 targets (*Map1b*, *Ppp1r9a*, *Sv2a*, *Syp*, and *Slc17a7/vGlut1*), showed increased polysome association in mutants. Celf4 target *Eif4a2* showed increased monosome association in mutants. *Dlg4/PSD95*, *Dmxl2*, *Gabra2* served as a negative control. Data normalized to *Actb* mRNA levels and presented as polysome/monosome ratios. Values represent the mean ± SEM of ≥5 technical replicates per target. *P* values are shown in Source data. Statistics: unpaired, two-tailed Welch's *t* test for parametric analysis, or Mann–Whitney test for non-parametric analysis. ns *p* > 0.05, *\**p* ≤ 0.05, **\**p* ≤ 0.01, **\*\***P* ≤ 0.0001. **b** ImmunoFISH showed strong co-localization of Celf4 protein (green) with *Eif4a2* mRNA (blue) and Eif4a2 protein (red) in the subplate (SP) of P0 neocortex (arrows). Arrowheads show weak Eif4a2 protein expression and absence of Celf4 protein, despite high *Eif4a2* mRNA levels. Sample size: *n* = 2 animals. Scale bar of 60x objective lens: 25 μm. Co-localization of Celf4 protein (green) with *Sv2a* mRNA (red) and *Dlg4/PSD95* mRNA (blue) in the SP of P0 neocortex (*n* = 2 male animals). Scale bar of 60x objective lens: 15 μm. **c** ImmunoFISH revealed a strong correlation of CELF4 protein expression (green) with *EIF4A2* mRNA (blue, right) and EIF4A2 protein (red) in the human SP at 19 PCW. DAPI is in blue. Scale bar of 60x objective lens: 13 μm. **d** ImmunoFISH showed co-labeling of CELF4 protein (green) and *SV2A* mRNA (blue) in the human SP at 19 PCW. EIF4A2 protein (green) colocalization with *SV2A* mRNA (blue) and SV2A protein (red) in the human SP at 19 PCW. EIF4A antibody was used for detection and imaging of CELF4-expressing neurons in the SP. Scale bar of 60x objective lens: 13 μm. **e** immunoFISH showed that Celf4 represses *Sv2a* mRNAs (red) translation and promotes *Eif4a2* mRNAs translation in mouse SP at P0. Eif4a2 protein levels (green and gray) are reduced, and Sv2a protein levels (blue and gray) are increased in the SP of *Emx1-Celf4* cKO (bottom) compared to the control (top). Middle and right: Quantification of Eif4a2 and Sv2a coexpression over total Eif4a2+ neurons (from **e**, left). Mutants showed decreased and increased percentages of Eif4a2$_{strong}$+Sv2a$_{weak}$ + SP neurons (middle) and Eif4a2$_{weak}$+Sv2a$_{strong}$ + SP neurons (right), respectively. Sample size: *n* = 3 male animals per genotype. Scale bar of 60x objective lens: 15 μm. Statistics: unpaired, two-tailed Welch's *t* test. *\**p* = 0.0066 ≤ 0.01, **\*\***p* < 0.0001. **f** Western blot revealed no difference in Sv2/Sv2a relative protein levels in whole neocortical homogenates of P0 WT and mutants (*n* = 3 male animals per genotype). Sv2/SV2A protein levels were increased in the mutant synaptoneurosomes (*n* = 4 independent synaptoneurosomal preps; 3 male neocortices pooled together as one biological sample). Western blots were run independently three times. Graphs were constructed by averaging individual Sv2 images normalized to multiple Gapdh exposure times. Data represent mean ± SEM. Statistics: unpaired, two-tailed Welch's *t* test for parametric analysis, or Mann–Whitney test for non-parametric analysis. ns *p* = 0.4101 > 0.05, **\*\***p* < 0.0001.

---

significantly decreased in female *Emx1-Cre* Celf4 cKO neonates (Fig. 5a, b). Similar observations were made for the individual number of VGAT + and Gphn+ puncta, their individual average size and total fluorescence intensities (Supplementary Fig. 11a, b). *VGAT* and *Gphn* mRNAs were not identified as Celf4 targets from RIP-RNAseq screen, suggesting that these changes reflect the secondary effect of synaptic development rather than a direct effect on these two markers.

We also detected significant alterations in thalamo-cortical glutamatergic synapses. In fact, the vGlut2 + PSD95+ overlapping area was significantly increased in the SP of male *Emx1-Celf4* cKO neonates compared to the control, while female mutant cortices remained unaltered (Fig. 5c, d). After measuring parameters for individual non-nuclear puncta in the SP area, we noticed a significant increase in the total number of vGlut2+ puncta in male mutants. In contrast, a significant decrease in the number, area and total intensities of individual vGlut2+ puncta were detected in the SP of *Emx1-Celf4* female cKOs (Supplementary Fig. 11c), suggestive of a potential increase in more immature synapses. Similar observations were made with the vGlut1 marker of cortico-cortical glutamatergic neurons. Significantly increased colocalization of non-nuclear vGlut1 + PSD95+ puncta was observed in the SP of *Emx1-Celf4* male cKOs (Figs. 5e, f). These results align with our previous findings that *vGlut1* mRNA is translationally derepressed in *Emx1-Celf4* cKO neocortex (Fig. 4a). However, all parameters for individual vGlut1 measurements remained unaffected in both male and female mutants compared to sex-matched control (Supplementary Fig. 11d). Quantification of individual PSD95+ puncta in the SP zone showed significant increase in their number only in male mutant cortices, while no significant changes were observed for *Emx1-Celf4* female cortices (Supplementary Fig. 11e). These results indicate that Celf4 is required to establish the proper balance of prenatal synaptic inputs to the developing cortex, and that sex influences this function.

This study underscores the opinion that early-onset dysfunctions of synapse pathways, collectively regarded as synaptopathies, are a major cause of NDDs[88,89]. To understand and characterize the synaptic involvement in the NDDs origin, it is important to first understand sex differences in unaffected individuals. This observation is supported by our findings in a mouse model which revealed significant baseline sex differences in both GABAergic and glutamatergic synapses. Our data also suggested that RBP Celf4 has a sexually dimorphic role in synaptic formation and composition during mouse neocortical development. One explanation might be that Celf4 binds to and translationally regulates its sex-specific target mRNAs to ultimately modulate the sex-specific synaptic output. Taken together, the prominent sex differences in GABAergic and glutamatergic synaptic markers observed upon *Celf4* loss may have important implications for understanding how synaptic deficits in ASD differ in males and females.

## Discussion

Although prenatal neocortical cells express a multitude of protein-coding transcripts that are stable at transcriptional level, the abundance of each transcript has little predictive value for estimating protein expression levels[18–21]. This mRNA-to-protein disparity is a common feature in other biological systems[90], emphasizing the critical need to better understand the spatiotemporal control of protein synthesis during nervous system development. As the complex steps underlying corticogenesis require precise spatiotemporal regulation, the specific mRNAs must be translated into proteins at the right pace, time and place. For the first time, this study has revealed the translational status of mRNAs associated with less translating 40S-60S-80S monosomes and actively-translating polysomes in human fetal neocortices at different developmental phases and in distinct cell subtypes. We identified CELF4/Celf4, an RBP strongly associated with NDDs and ASD, as the translational regulator of prenatal synaptic development in the SP. Our findings enhance the current knowledge on tissue-specific cellular heterogeneity and the evolutionary advancement of neocortical cellular biology, particularly in the synapse-rich SP, the most prominent developmental compartment in the primate prenatal neocortex.

Standard RNAseq based on polysome fractionation analysis is an effective and well-established method to monitor the translational status of mRNAs but is also recognized to have some limitations. Arguably, choice of normalization method by which experimental variations are corrected can have significant impact on the downstream analysis results. For example, our attempt to normalize polysome-associated transcripts with free mRNA abundance resulted in the increased signal-to-noise ratio, masking any genes that could be declared as significantly different. In contrast, filtering out those transcripts with major changes at the transcript level (either due to changes in the transcription itself or mRNA stability) enabled identification of relatively unchanged mRNAs whose expression may be regulated by the translational control mechanisms. Our study revealed a significant number of transcripts that underwent changes in their translational status, with an approximately equal number being either translationally repressed or derepressed (Fig. 1e), from early fetal (11/12 PCW) to early midfetal (14/15 PCW) phases. The KEGG pathway analysis

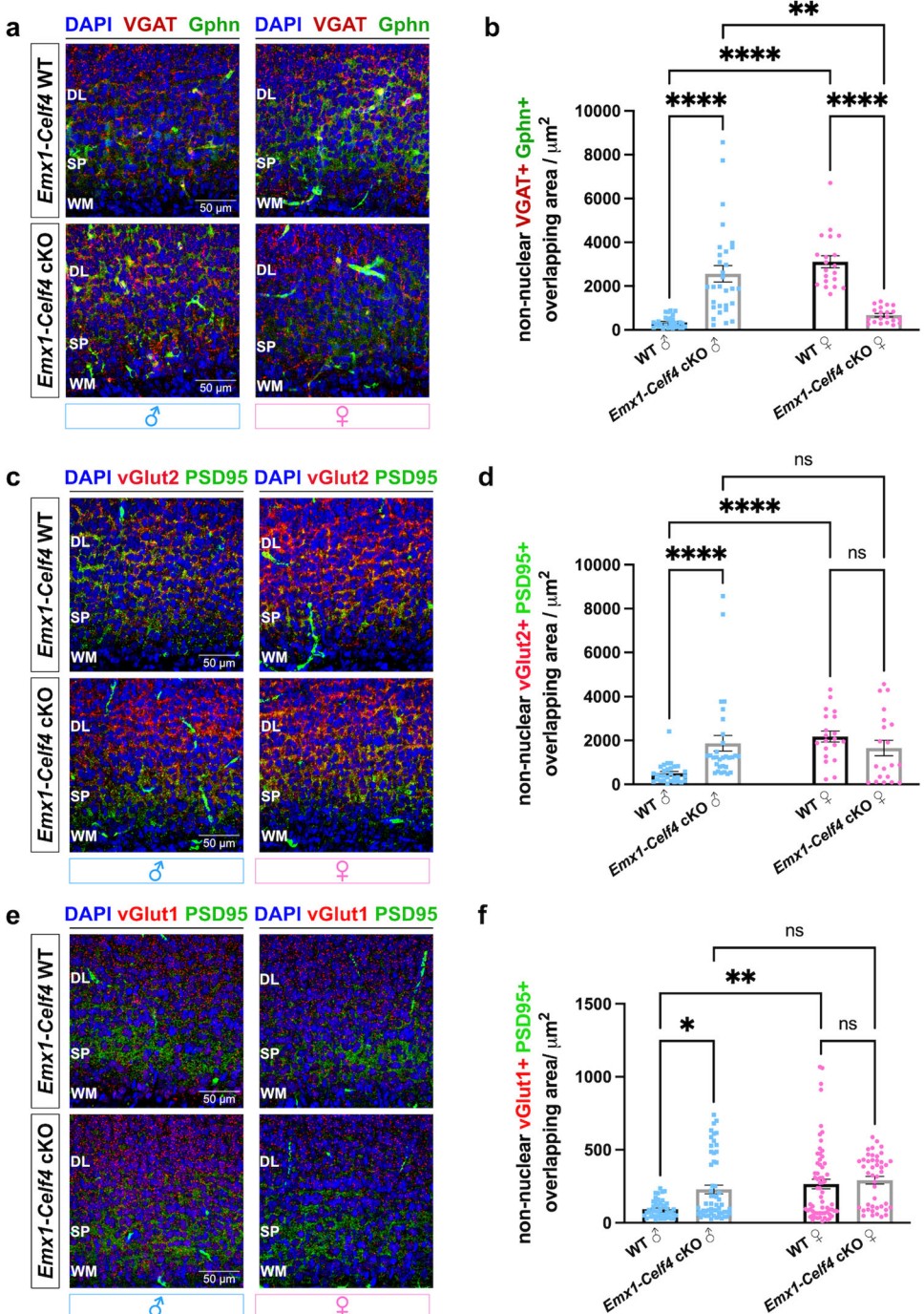

**Fig. 5 | Celf4 modulates the formation of prenatal GABAergic, thalamo-cortical and cortico-cortical glutamatergic synapses in sex-specific fashion.** Representative confocal images of subplate (SP) at P0 stained for (**a**) pre-synaptic (VGAT, red) and post-synaptic (Gphn, green) GABAergic markers (*n* = 3 male animals per genotype, *n* = 2 female animals per genotype), (**c**) pre-synaptic (vGlut2, red) and post-synaptic (PSD95, green) glutamatergic markers (*n* = 3 male animals per genotype, *n* = 2 female animals per genotype), and (**e**) pre-synaptic (vGlut1, red) and post-synaptic (PSD95, green) glutamatergic markers (*n* = 3–4 male and female animals per sex and genotype). DAPI positive nuclei are in blue. Location of the SP was previously confirmed with the established SP marker, Complexin 3.

**b**, **d**, **f** Quantification of overlapping puncta between corresponding pre- and post-synaptic pairs in the SP of P0 WT and *Emx1-Celf4* cKOs, separated by the sex (>1000 puncta/image; analyzed 20–57 images per sex/genotype). Sample size: *n* = 3 male animals per genotype, *n* = 2 female animals per genotype for VGAT/Gphn and vGlut2/PSD95 analyses; *n* = 3–4 male and female animals per sex and genotype for vGlut1/PSD95 analysis. Data represent mean of non-nuclear mask of overlap at the cell level ± SEM. Statistics: Kruskal-Wallis non-parametric test with Dunn's post-hoc test for multiple comparisons (*P* values shown in Source data). ns *p* > 0.05, **p* ≤ 0.05, ***p* ≤ 0.01, *****p* ≤ 0.0001. Scale bar of 60x objective lens: 50 μm. Deep layers, White matter, WM.

showed that the group of translationally derepressed mRNAs during early-to-late transition was involved in several pathways, including the "spliceosome." This finding is in agreement with previous reports on the importance of neuronal splicing programs during neocortical development[25,91]. This remarkable developmental communication between splicing and regulated translation amplifies the ultimate proteomic diversity from relatively limited number of protein coding genes.

The analysis of snRNAseq data also presents some challenges, especially when estimating differential expression of each gene, or the lack thereof, at often sparse and noisy single nucleus resolution. While this limitation needs to be improved, snRNAseq technology is still the most commonly used approach to analyze the transcript expression of single cells. In this study, we compared the cellular transcriptomes with translatomes from prenatal human frontal cortex (Fig. 1f). Interestingly, fewer mRNAs with stable transcription levels showed a shift in translational status from early midfetal (14/15 PCW) to late midfetal (17/18 PCW) phases, implying a larger combinatorial set of translational output during earlier phases of cortical development. Translationally regulated mRNAs from mid to late phases showed almost no overlap with early-to-mid mRNAs regulated at the translational level (Fig. 1e). For example, the "spliceosome" group of mRNAs exhibited translational derepression at the mid phase (Supplementary Fig. 12a, left and right) but became translationally repressed in the late phase (Supplementary Fig. 12a, left and right). In contrast, the synapse-associated mRNAs were first repressed in the mid phase (Supplementary Fig. 12a, right), then translationally derepressed in the last phase (Supplementary Fig. 12a, right). These data suggest that translational regulation decelerates as the neocortex develops, indicating the existence of distinct translational regulators that govern protein expression signatures specific to each developmental phase and cell type in human fetal cortices. This, in turn, enables the remarkably enhanced diversity and complexity of human fetal neocortex.

Translational reprograming appears to be more pronounced in deep layer neurons and evolutionarily advanced SP, suggesting that cell-type specific RBPs may play a role in modulating this process. In particular, the RBP CELF4/Celf4 is highly expressed in the developing SP and excitatory layer 5/6 neurons (Figs. 2b and 3a), where it exerts post-transcriptional regulatory functions necessary for proper human and mouse neocortical development. We observed that CELF4 has a tendency to bind and regulate mRNAs that undergo translational derepression during the course of neocortical and SP development (Supplementary Fig. 12b). Furthermore, we found that CELF4 targets more mRNAs in the mid phase than in the late phase, suggesting a possible "slowdown" of its function during development or the potential involvement of other RBPs outcompeting CELF4 and taking over the control of certain mRNAs. When we analyzed the translational status of human CELF4 targets in *Emx1-Celf4* cKOs (Fig. 3d), we found that Celf4/CELF4 primarily controls the translational repression of specific presynaptic mRNAs, like *Syp* and *Sv2a* (Supplementary Fig. 9c). Both Celf4/CELF4 protein and presynaptic mRNAs/proteins are expressed in the synapse-rich SP area during mouse and human corticogenesis (Fig. 4). Importantly, we identified that Celf4/CELF4 targets some of the same orthologs between mouse and human developing cortices (Fig. 3c), suggesting that Celf4/CELF4 may have a conserved role in forming and/or maintaining prenatal neocortical synapses.

In addition, we discovered a large group of 2352 CELF4 targets that are unique to humans and displayed phase-specific expression and translational regulation. As expected, these human-specific targets were enriched in various synaptic functions (e.g. "regulation of synaptic vesicle exocytosis", "dendrite morphogenesis" and "regulation of synaptic plasticity"). The 14/15 PCW phases, during which CELF4 binds to the highest number of translationally regulated targets, are particularly important in the development of the SP. These phases are characterized by a tremendous expansion of the SP (called "the second cortical plate"), and a concurrent increase in synapses due to the ingrowth of major thalamocortical and basal forebrain afferents[14,16,17]. This suggests that CELF4, at least partially, participates in these critical developmental dynamics.

An unbiased translational screen of *Emx1-Cre Celf4* cKO cortices revealed that the majority of Celf4 target mRNAs were translationally derepressed, while non-Celf4 targets showed nearly equal degree of translational repression and derepression (data not shown).

Consistent with these findings, we found that two Celf4 target mRNAs, *Syp* and *Sv2a*, were translationally derepressed in *Emx1-Cre* Celf4 cKOs, as evidenced by their increased protein expression in the synapse-rich SP of the mutant developing cortices (Fig. 4e and Supplementary Fig. 10d). These findings further confirm translational repression as one of the major cellular and molecular mechanisms in the developing brain[18–21]. Moreover, our discovery of a significant overlap in a distinct group of target mRNAs between mouse and human developing cortices indicates that Celf4/CELF4 may contribute to the evolutionarily conserved initial prenatal synapse development.

Finally, and to our surprise, we found that Celf4 regulates prenatal GABAergic and glutamatergic synapses in a sex-specific fashion at P0 developmental stage, possibly contributing to the sex prevalence in a number of NDDs, including ASD. Future studies should examine if sex-dependent synaptic differences are present at other prenatal and postnatal stages, and whether Celf4 regulates the expression of other synaptic markers during neocortical development. Consistent with the synaptic deficit phenotype (Fig. 5) and previously published data[79,80], we observed that *Celf4* loss causes occasional hypo- and hyperactivity and spontaneous recurrent seizures later on, during mouse postanal and adult stages (not shown). Our GO enrichment analyses (Fig. 3g) and RIP-RNA seq data (Figs. 2f, 3c) suggest that early prenatal *Celf4* loss from excitatory neurons may result in various synaptic defects in the SP, potentially impairing cortical excitability postnatally and causing epilepsy and depression[79,80,92]. Moreover, the identified sex-specific roles of Celf4 in mice suggest that the translational control may be the point of sex-dependent vulnerability in the number of NDDs, including the ASD. Overall, these data indicate that Celf4 acts as an important translational regulator in both mouse and human, with cell subtype specific functions in prenatal neocortices and possibly in a sex-specific manner.

Collectively, the precise spatiotemporal control of mRNA fate at a cell-subtype level has a critical effect on mammalian brain development and is evolutionarily conserved from mice to human. However, the exact coordination of these events by sex- and cell-type specific translational regulators remains unclear. Therefore, future studies should focus on investigating sex- and cell-type specific translational regulators to shed light on the differences in NDD pathogenesis between males and females. The revelation of sex- and cell-type specific patterns of translational repression and derepression in developing cells holds great promise for future research efforts related to NDDs.

## Methods

### Animals and approvals

Animal care and experimental procedures involving animals were performed in accordance with the guidelines established by Rutgers-RWJMS Institutional Animal Care and Use Committee (protocol: I12-065). To investigate the neurodevelopmental role of Celf4 in vivo, we obtained Celf4f/f mice (strain name: B6.129-*Celf4tm1.1Frk*/Frk, Jax strain: 018126) from Jackson Laboratory. Wild type (WT) littermate control and *Celf4* conditional-deletion animals were generated by crossing Celf4f/f mice with *Emx1*-Cre mice (strain name: B6.129S2-*Emx1tm1(cre)Krj*/J, Jax strain: 005628) transgenic line that drives the expression of Cre recombinase in the cortical progenitors and postmitotic neurons of neocortex. For timed pregnancies in which the day of vaginal plug detection was considered as embryonic day 0 (E0), we used WT littermate control and *Emx1-Celf4* cKOs. Mice were kept on a 12:12-h light/dark cycle (lights on at 7:00 am), and received water and food ad libitum at room temperature and 40-60% humidity. Mice of either gender and genotype (WT and cKO) were used in all mouse-related experiments.

### Postmortem human brain tissue

All experiments on human tissue were carried out following the Declaration of Helsinki 2000. Tissue was collected with appropriate

maternal consent, along with approval for use in research. Ethical approval was obtained from therelevant ethics committees. The tissue sampling was approved by the Internal Review Board of the Ethical Committee of the University of Zagreb, School of Medicine. Postmortem human fetal brain samples from 11 to 21 post-conceptual weeks (PCW) were immersion-fixed in 4% paraformaldehyde (PFA, Biognost, no. FNB4) in 0.1 M phosphate buffered saline (PBS; pH = 7.4) within the 6 hours of postmortem time. Age of the brain specimens in PCW was determined by crown-rump length (CRL) and information from the pregnancy records. Sex was not determined in human fetal tissue. Part of the human embryonic and fetal material was provided by the Joint MRC / Wellcome Trust (grant # MR/006237/1 and #099175/Z/ 12/Z ) Human Developmental Biology Resource.

## Polysome fractionation

For polysome profiling, we used human fresh frozen neocortices from 11 to 20 PCW ($n = 2$ spins with independent neocortex per developmental phase). Mouse embryonic neocortices at postnatal day 0 (P0) were dissected in ice-cold HBSS media supplemented with 5 mg/ml glucose and 20 mM HEPES (pH=7.2), followed by instant freezing on dry ice. Three mouse neocortices were pooled together and used for single spin ($n = 3$ spins per genotype; WT versus *Emx1-Celf4* cKO). Prior polysome fractionation, human and mouse tissue was homogenized in freshly prepared polysome extraction buffer (PEB) supplemented with EDTA-free protease inhibitor (Santa Cruz Biotechnology; no. sc-29131), RNaseOut (Invitrogen; no. 100000840), 20 mM DTT (Invitrogen; no. NP0009), and 0.1 mg/mL cyclohexamide (Santa Cruz Biotechnology; no. sc-3508A). Lysis was performed by pipetting gently up and down on ice for 15 minutes. Lysate was then centrifuged at 5000 rpm ($2040 \times g$) for 10 min at 4 °C, and the resultant supernatant was spun at 14,000 rpm ($16,000 \times g$) for 5 min at 4 °C. Following sample measurement on Nanodrop ($A_{260}$) and normalization of the total loading to total mRNA content, prepared neocortical input (120 μg) was then layered on to 10–50% sucrose density gradient in thin wall polypropylene centrifuge tubes (no. 347357). A 25% of the input was stored for further processing. For the optimal quality, gradients were poured in 2 ml polyallomer tubes (Beckman Coulter; no. 347357) at the night before the centrifugation and allowed to stabilize overnight at 4 °C. Next, gradients were spun at 4 °C and 39,000 rpm (-130,140 × $g$) for 120 min using Thermo Fisher Sorvall MX 120+ micro-ultracentrifuge and the Sorvall S55-S swinging bucket rotor. Polysome profiles were generated with a continuous flow of Fluorinert FC-40 (Sigma; no. F9755) and $A_{254}$ recording using a Brandel fraction collector (no. 621140007) with UV absorbance recorder (Brandel UA-6). Samples were then frozen at −80 °C until further use. The data were digitally recorded using WinDaq acquisition software (version 2.84). Individual gradient fractions were aligned with respective profiles and equal fraction volumes were pooled as 40S-60S-80S monosomes (fraction no. 6, 7, 8, 9) for isolation of 40S−60S−80S-associated cytoplasmic RNA and polysomes (fraction no. 11, 12, 13, 14, 15) for isolation of polysome-associated cytoplasmic RNA. Isolated RNAs were subjected to RNA-seq transcriptome analysis and qRT-PCR analysis for gene expression profiles assessment.

## RNA immunoprecipitation (RIP)

E17 mouse neocortices were dissected, pooled ($n = 6$–9 neocortices), and processed as single biological sample. A total of five biological samples were used for RNA immunoprecipitation (RIP) experiments. For human RIP experiments, we used fresh frozen tissue of human fetal frontal cortex at 11 and 12 PCW (total sample size $n = 3$), 14 and 15 PCW (total sample size $n = 2$) and 17, 18 and 20 PCW (total sample size $n = 3$). RIP was performed using the EZ-Magna RIP RNA-Binding Protein Immunoprecipitation Kit (Millipore, no. 17-701) following the manufacturer's instructions. Briefly, 50 μl of magnetic beads were incubated with either 2.5 μg of CELF4 antibody (Invitrogen, no. PA5-58196) or

2.5 μg of a negative control IgG rabbit antibody (provided in the RIP kit) in 500 μl of wash buffer for 30 min to prepare magnetic beads for immunoprecipitation. To immunoprecipitate Celf4/CELF4−RNA complexes, the antibody-bounds beads were incubated with 100 μl cell lysate in 900 μl immunoprecipitation buffer overnight at 4 °C with rotation. The beads containing RNA-binding protein-RNA complexes were then washed a total of six times with 500 μl of cold wash buffer. To purify RNA and digest the proteins, Celf4/CELF4 immunoprecipitates were resuspended in 150 μl of proteinase K buffer and incubated at 55 °C for 30 min with shaking. Afterwards, the RNA was purified with the addition of phenol:chloroform: isoamyl alcohol (125:24:1, pH = 4.3), Salt Solution I, Salt Solution II, Precipitate Enhancer and absolute ethanol. Upon RNA precipitation, approximately 100 ng of coimmunoprecipitated RNA was treated with the TURBO DNA-free Kit (Invitrogen, no. AM1907) following the manufacturer's specifications. Finally, RNA was used in qRT-PCR and RNA-Seq analysis; the latter was carried out using triplicate samples. The qRT-PCR was performed in at least four technical replicates. Fold changes were calculated using the $2−(\Delta\Delta Ct)$ method, and GAPDH mRNA was used for normalization. To discriminate the specific binding of Celf4/CELF4 to target mRNAs, each target mRNA was normalized based on ΔCt of IgG, which served as a negative control.

## RNA isolation

Total RNA was extracted either from pooled polysome profiling fractions, RIP samples or total neocortices using TRIzol LS (Life Technologies no. 10296028) in a ratio 3:1 of Trizol LS:sample, mostly following the manufacturer's specifications. Then, chloroform was used twice to ensure proper separation of organic phases and isolation of the aqueous phase. Precipitation of total RNA was achieved using one volume of isopropanol/glycogen (Thermo Fisher; no. R0551). The pellet was washed with 75% ethanol, air-dried and resuspended in Molecular Biology Grade water (Corning; no. 46-000-CV) supplemented with RNAseOut (for qRT-PCR analysis). Resolubilizing the RNA pellet after precipitation lasted for approximately 30 min with frequent vortexing. The residual genomic DNAs were removed by subjecting the mRNAs to DNase treatment according to the manufacturer's protocol (Invitrogen; no. AM1907) with slight modifications: DNase was either inactivated as per manufacturer's specifications or with phenol:chloroform (Fisher Scientific; no. BP1754I), followed by chloroform extraction and precipitation with 3 M sodium acetate (G-Biosciences, no. R010). Samples were stored at −80 °C in 100% ethanol until further processing. RNA was dissolved in 20ul of Molecular Biology Grade (MBG) water for RNA sequencing purposes, or in MBG water supplemented with RNAseOUT for qRT-PCR analysis.

## Single-nucleus RNA sequencing (snRNAseq)

All postmortem human fetal brain samples at 11, 12, 14, 15, 17 and 18 post-conceptual weeks (PCW) were stored at −80 °C until use (submitted in duplicates). Single-nucleus capture (sample coverage of ~10,000 nuclei/sample) and library preparation were performed by Azenta, Inc. (South Plainfield, NJ). RNAseq libraries were prepared using the 10X Genomics® Chromium™ 3′ gene expression workflow, which is based on Drop-Seq technology. The single-nucleus libraries were sequenced to a depth of ~100,000 reads/nucleus on the Illumina® NovaSeq™6000 at Azenta, Inc. Chromium single cell data was processed through the Cell Ranger pipeline (10X Genomics) for demultiplexing of raw sequencing libraries into FASTQ files for each sample and conversion into a UMI raw count matrix. Using the single cell R package Seurat (version 4)[93], the matrices from individual samples were loaded into a single Seurat object, clustered by a shared nearest neighbor (SNN) algorithm, colored by SNN cluster assignment using Seurat FindClusters and visualized per cell type in the two-dimensional space using scCustomize v.1.1.1[94]. The manual annotation of the cell clusters was done using a predefined set of established marker genes:

*TOP2A* and *MIK67* for cortical progenitors; *PAX6, GLI3, PDGFD, ZFHX4, HES1, PTN,* SLC1A3, *ID4* for aRG; *FBN2, HS6ST2, PTN, TNC, LIFR MOXD1* and *PTPRZ1* for bRG; *EOMES* and *HES6* for IP; *RELN* and *NDNF* for CR cells; *DPP10, ST18, TMEM178A, SLIT3, NR4A2, CDH18, TLE4, MGST1, LMO3, PDE1A, NRCAM, FBXW7, LMO7, NPR3, PLEKHA5, SORCS1, SLC4A10, GPR2* and *PCP4* for SP neurons; *UNC5D, SATB2, PRSS12, POU3F2, CUX1, CUX2, TLE3, MEF2C, FOXP1, POU3F3, MDGA1, RORB, LHX2, LMO4, TLE1, NWD2* and *DOK5* for ExN L2-4; *SOX5, TLE4, BCL11B, LMO3, SEMA3A, CRYM, LDB2, FOXP2, TOX, ETV1, LMO7, HS3ST4, MARCKSL1* and *RPRM* for ExN L5/6; *LRRC4C, TNR, BCAN, SMOC1,PDG-FRA, NOVA1, PTN PCDH15* for OPC; *BCAS1, MBP* and *PCDH9* for Oligo; *ADGRV1, SLC1A3, QKI, HES1, PTPRZ1, MEIS2, ATP1A2, GLUL* and *HES5* for Astro; *NXPH1, ADARB2, GAD1, GAD2, NPAS1* and *ARX* for InN; *ERBB4, LHX6, RELN, SST, MAF* and *MAFB* for MGE InN; *NR2F1, NR2F2, CCK, CALB2* and *CCDC88A* for CGE InN; *IGFBP7, CLDN5, COL3A1, H3F3B, PECAM1, MYO1E, FLT1,* and *COBLL1* for Endo; *MYO1E, FLT1, COBLL1, FN1, LAMA4, LAMA2, COL1A1, COL4A2* and *CALD1* for Peri; SPP1, *NEAT1, SAT1, RGS1, P2RY12, ZFP36L1, SLCO2B1, FTL, A2M* and *CSF1R* for Micro.

Differential expression between clusters was calculated with Seurat's FindAllMarkers() function (Wilcoxon rank-sum test) using default settings, with p-values adjusted based on the Bonferroni correction. pct.1 indicates the percent of nuclei that express the gene in the target cluster, pct.2 indicates the percent of non-target nuclei that express the gene. Genes were considered enriched if they were detected in at least 10% of cells in the cluster, 0.25 $\log_2$ fold enriched, and Bonferroni corrected *p* value < 0.05 (Supplementary Data 1).

To identify the top-ranked RNA-binding protein associated with the "synapt" annotations, the same criteria were applied except for the percent cutoff, which was set at 0.3 (30%) (Supplementary Data 7). The DEG analysis was conducted by comparing SP clusters against all other neuronal cluster, individually for early (SP: E0, E7, E9 versus Neuronal: E2, E4, E5, E6, E8, E16), mid (SP: M5, M13 versus Neuronal: M0, M1, M2, M3, M7, M8, M9, M10, M11, M12) and late (SP: L4, L16, L15 versus Neuronal: L0, L1, L2, L3, L5, L6, L7, L10, L13, L14, L18).

For intersection analyses, we used minimum mean expression by cluster (>0.1) with at least 30% of cells in a cluster expressing the gene. Enrichment significance was defined by hypergeometric test, followed by *p* value adjustment using the BH-procedure; significance was set at *p* < 0.05. All sequencing data have been deposited in the Gene Expression Omnibus (http://www.ncbi.nlm.nih.gov/projects/geo/) and can be accessed online under SuperSeries number GSE214534.

## Standard RNA sequencing (RNAseq) analysis

The RNA isolated from mouse and human RIP-RNAseq data, as well as input, monosome and polysome fractions obtained using either human fetal neocortices (from 11 to 20 PCWs) or mouse neocortices were stored at −80 °C until further use. Human RNA samples were submitted as two biological replicates, whereas mouse RNA samples were submitted in triplicates per each sample. Quality control testing, poly(A)-enriched sequencing library preparations, and Illumina®Hi-Seq® (configuration: 2x150bp) sequencing was performed by Azenta, Inc. (South Plainfield, NJ). Fastq files for individual sample were trimmed to remove primer sequences with Trimmomatic (v.0.3.6)[95]. Reads were then mapped to either human genome (hg38/GRCh38 assembly) or mouse genome (GRCm39 assembly) using STAR (v2.5.2b) and a table of counts per gene was extracted with the featureCounts() function from the Subread package (v1.5.2)[96]. Differential gene expression was modeled in DESeq2[97] with filters of two-fold changes and adjusted *p* value of 0.05 unless otherwise stated. For isoform (splice variant) mapping, fastq files were trimmed and filtered with fastp (v0.12.2)[98] and mapped to a library of transcripts using Kallisto (v0.46.0)[99]. Tables of transcript counts per sample were fit to a DESeq2 model. Two-tailed Wald test was used to identify DEGs in pairwise comparisons in DESeq2. *P* values were corrected using a false discovery rate (FDR) of 5% according to the Benjamini-Hochberg method.

Functional analysis was performed on the statistically significant set of genes by using either the clusterProfiler package (v3.10.1)[59] in R or PANTHER[69] website (Protein Analysis Through Evolutionary Relationships, http://pantherdb.org). All sequencing data have been deposited in the Gene Expression Omnibus (http://www.ncbi.nlm.nih.gov/projects/geo/) and can be accessed online under SuperSeries number GSE214534.

## Quantitative real-time PCR (qRT-PCR)

As described in the relevant methods subsection, mRNAs were isolated using TRIzol LS Reagent (Invitrogen, no. 10296028), solubilized in 20 µL of MBG water and reverse-transcribed to single strand cDNA using M-MLV Reverse Transcriptase (Promega, no. M170B) reaction. PowerUp SYBR Green Master Mix (Applied Biosystems, no. A25742) and target-specific primers (designed with the Primer-BLAST and listed in Supplementary Data 11) were used to perform RT-PCR reactions in QuantStudio3 Real-Time PCR machine (Thermo Fisher Scientific), as per manufacturer guidelines. Amplification reactions were conducted under the following conditions: 10 min at 95 °C, 40 cycles of 95 °C for 15 s, 55 °C for 15 s, and 72 °C for 30 s. All quantitative measurements were analyzed using the ΔΔCt method and normalized to the housekeeping gene control glyceraldehyde-3-phosphatedehydrogenase (*Gapdh*), unless otherwise specified. Non-template negative controls were used to verify amplification and determine background levels of amplification. For polysome profiling experiment, each target gene's mean expression levels in the monosome and polysome fraction was normalized to the *β-actin*, respectively, and a polysome/monosome ratio (ΔΔΔCt [ΔΔCt (ΔCt Poly - ΔCt Mono) − Average (ΔΔCt WT)]) was calculated using parametric unpaired, two-tailed Welch's *t* test or non-parametric unpaired, two-tailed Mann–Whitney test.

## Enrichment analyses

Enrichment analyses were performed using permutations in R custom script (R studio 1.2.5042). We defined three sets of gene list: input gene sets, including the genes representative of each cell cluster; target gene sets, including risk genes for ASD and severe NDDs; and the background gene set, including all genes detected in the snRNAseq data and with scores of association with ASD and/or severe NDDs. First, the input gene sets were defined as the set of genes whose mRNAs have a $\log_2$fold-change (FC) > 0.5849 in each cluster identified in each developmental phase. This represents the genes that have at least 1.5 higher expression in the given cell cluster compared to all other cell clusters in the defined developmental time point. Second, two target gene sets were extracted. The ASD risk genes set includes 183 autosomal genes with genome-wide significance association (FDR < 0.05) in Fu et al (2022)[43]. The severe NDD risk genes set (DDG2P) includes 1,337 unique genes with the "brain" term included in "organ specificity" in the Developmental Disorders Genotype-Phenotype Database (DDG2P, https://www.deciphergenomics.org/ddd/ddgenes, June 2022)[54]. Third, the background gene list was defined as the list of genes encoding mRNAs detected in each developmental phase (early, mid, and late) and was also listed as genes with annotated statistical scores of association with ASD in Fu et al (2022)[43] (18,128 "TADA" genes listed in Supplemental Table 9 of the original paper). For example, 6,789 of the unique 8207 genes expressed in all clusters of the early time point overlap with the 18,128 "TADA" genes in Fu et al. (2022)[43], so the 6789 genes are used as a background list for the enrichment analyses of the early clusters. For each comparison between the input and target gene sets, we first constructed the empirical distribution by sampling the same number of genes as in the input gene set from the background gene set 10,000 times. The *P* value was then computed by calculating the number of sampled gene lists that had at least as many overlapping genes with the target gene sets as the input gene set, divided by 10,000 iterations.

## Preparation of the synaptoneurosomes (SNs)

P0 WT and *Emx1-Celf4* cKO neocortices of either sex were dissected as described in the Polysome fractionation subsection. Cortical SNs were generated based on a previously published protocol[100]. Briefly, three frozen neocortices (*n* = 4 per condition) were pooled together and processed through the cold homogenization in 1 ml of freshly prepared homogenization buffer (0.32 M sucrose, 1 mM EDTA, 1 mg/ mL BSA, 5 mM HEPES, pH=7.4) in a glass-Teflon® douncer. After centrifugating twice, pelleted SNs were resuspended in 550 μL of freshly prepared Krebs Ringer buffer (140 mM NaCl, 5 mM KCl, 5 mM glucose, 1 mM EDTA, 10 mM HEPES, pH=7.4). Next, SNs were isolated by addition of 450 μL of Percoll™ (final 45% v/v), followed by centrifugation at 14,000 rpm (16,000 × *g*) for 2 min at 4 °C to separate the flotation gradient. The pelleted SNs were washed once in 1 mL of Krebs-Ringer buffer, centrifuged at 14,000 rpm (16,000 × *g*) for 30 s at 4 °C, and resuspended in 50 ml RIPA lysis and extraction buffer (Thermo Fisher Scientific, no. 89900) supplemented with EDTA-free protease inhibitor (Santa Cruz Biotechnology). Protein concentration was measured using the Pierce bicinchoninic acid assay (Thermo Fisher Scientific, no. 23227), and relative abundance of specific proteins was analyzed by Western blot analysis.

## Western blotting (WB)

Frozen neocortical tissue from individual P0 WT and *Emx1-Celf4* cKO mice of either sex (*n* = 3 neocortices per condition) was lysed in 20 μL Tissue Protein Extraction Reagent (Thermo Scientific, no. 78510) supplemented with EDTA-free protease inhibitor (Santa Cruz Biotechnology) at 4 °C. Synaptoneurosomal tissue was lysed as described in the relevant methods subsection. After centrifugation for 10,000 rpm (8160 × *g*) at 4 °C for 10 min, the supernatant was collected for quantification of the protein concentrations on a Nanodrop ND-1000 spectrophotomer using the Pierce 660 nm reagent (Thermo Fisher, no. 22660), unless otherwise noted. Typically, 3.2 μg of total proteins were loaded for each sample and analyzed using the NuPAGE system (Life Technologies) with 4–12% Bis-Tris gels (no. NP0335/6). Separated proteins were transferred to nitrocellulose membranes (GVS Life Sciences, No. 1215471), which were then blocked for 3 h at room temperature with 5% nonfat dried milk (VWR, no. M203) supplemented with 10% fetal bovine serum (FBS) (Gemini, 900-108) in PBS with 0.4% Triton-X-100 (PBS-T). The blots were incubated in the primary antibody solution diluted in 10% FBS in PBS-T overnight at 4 °C with gentle rotation. The next morning, the blots were washed three times in PBS-T for 10 min and incubated at room temperature for 1 h in corresponding secondary antibody solution diluted in 10% FBS in PBS-T. The ChemiGlow West Chemiluminescence Substrate Kit (Proteinsimple, no. 60-12596-00-2) was used for protein detection, followed by signal visualization with Azure 600® (Azure Byosistems, no. AZI600). Sv2 and Gapdh antibodies used for immunoblot detection are listed in the corresponding methods subsection. Band quantification was performed by densitometry using ImageJ2 software (v2.3.0/1.53 f; Rasband, W.S., ImageJ, U. S. National Institutes of Health, https://imagej.net/software/imagej2/). The protein of interest was normalized to Gapdh levels on the same blot according to the following procedure: multiple images of Sv2 protein at different exposure times were normalized to multiple images with different exposure times of the housekeeping protein, Gapdh, from the same blot. The normalized values from individual Sv2 image at several Gapdh exposure times were averaged and used as single data point. All Western blots were performed with two to three technical repeats, and parametric unpaired, two-tailed Welch's *t* test or non-parametric unpaired, two-tailed Mann-Whitney test were used to compare protein relative abundances between two conditions.

## Immuno-electron microscopy (immuno-EM)

To collect and dissect mouse embryonic neocortices at E15 and E17, adult pregnant female CD-1 mice were purchased from Charles River Laboratories. E15 (*n* = 3) and E17 (*n* = 2) neocortices were fixed at 4 °C in ice-cold fixative containing 4% PFA (Sigma-Aldrich, no. 158127), and 0.05% glutaraldehyde in 0.1 M phosphate buffer (PB), pH 7.4, for 4 h. After fixation, brains were washed thoroughly in 0.1 M PB and stored in filtered 30% sucrose in 0.1 M PB until further processing. Coronal vibratome sections (70 μm) were sectioned using a Leica vibratome (Leica, no. VT1000 S) and collected in 0.1 M PB at room temperature.

Postmortem human prenatal brain tissue (prospective temporal cortex) at the age of 19 PCW was fixed in ice-cold 4% PFA and 0.05% glutaraldehyde for up to 24 h. The brain hemispheres were stored in 30% sucrose in 0.1 M PBS at 4 °C until further processing. For iEM immunostaining, subplate area of the temporal cortex (approximately 1 cm from pial surface) was dissected out, washed in 1x PBS, embedded in 3.2% agar and cut on vibratome at 80–100 μm thickness. Slices were then infiltrated with 30% sucrose in glass vials, followed by freezing on dry ice and rapid thawing; this process was repeated three times to achieve adequate antibody penetration and preservation of the tissue ultrastructure. To block endogenous peroxidase, human slices were immersed in 1% $H_2O_2$ in 0.1 M PB, then washed twice in 0.1 M PB for 10 min at RT.

Mouse and human free-floating sections were preincubated for 3 h at room temperature with gently rotating in blocking solution that consisted out of normal donkey serum (Jackson ImmunoResearch, no. 017-000-121), albumin (Biomatik, no. A2134), 0.2% Glycine (BDH, no. BDH4156), 0.2% L-lysine (Sigma, no. L5501), and 0.4%Triton (Sigma, no. X-100). Then, mouse brain sections were incubated with primary anti-Celf4 antibody (anti-BRUNOL4, Invitrogen, 1:500, no. PA5-58196) overnight at 4 °C with gentle shaking. Human brain sections were incubated with primary anti-Celf4 antibody (anti-BRUNOL4, Invitrogen, 1:300, no. PA5-58196) for two days at 4 °C with gentle shaking. After three washes for 10 min in 0.1 M PB, sections were incubated in biotinylated donkey anti-rabbit IgG secondary antibody (1/250, Jackson ImmunoResearch) diluted in a blocking solution with 0.04% Triton. Sections were washed three times in 0.1 M PB, incubated at room temperature for 1.5 hours (mouse sections) or 2.5 hours (human brain sections) with Avidin-biotin reagent and washed again three times for 5 min in 0.1 M PB. The peroxidase reaction was developed using the DAB-Nickle Peroxidase Substrate Kit (Vector Laboratories, no. SK-4100) for 5 min according to the manufacturer's instructions. The sections were then washed twice for 5 min in 0.1 M PB and post-fixed with 1 % OsO4 in 0.1 M PB for 10 min in darkness. After three washes for 5 min in 0.1 M PB, neocortical sections were dehydrated for 5 min in 50% (v/v) ethanol, 5 min in 75% (v/v) ethanol, 5 min in 95% (v/v) ethanol, and three times in 100% (v/v) ethanol. Ethanol was then cleared from the tissue twice with acetone and once with acetone mixed with epoxy resin. The sections were then flat embedded in epoxy resin and polymerized at 65 °C for 48 hours. The next day, thin sections of cold interference color were made using a Leica ultramicrotome (Leica Microsystems, Wetzlar, Germany) and collected on copper grids for imaging. All sections were photographed using Philips CM12 electron microscope, operating at 80 kV, and equipped with AMT-XR11 digital camera (magnification of 8000x).

## Immunofluorescence (IF), Fluorescence in situ hybridization (FISH) and imaging of human brain tissue

Tissue was fixed in 4% PFA (Biognost, no. FNB4) for up to 48 h, dissected coronally in three blocks, embedded in paraffin (Merck, no. 107300) and sectioned (10 μm thick sections) on a microtome (Leica, SM2000R). Fixed coronal sections were then mounted on slides. Prior to immunohistochemistry, a standard process of deparaffinization was performed in a series of xylol and alcohol. After four washes in 1x PBS,

the sections were incubated in blocking solution containing 1% BSA and 0.5% Triton X-100 in PBS for 1 hour. Blocking solution was then replaced with primary antibody solution (rabbit anti-CELF4, 1:150; rabbit anti-SV2A, 1:400; mouse anti-EIF4A2, 1:400; mouse anti-SYP, 1:400) which was diluted in blocking solutions and kept overnight at 4 °C. The next day, the sections were washed three times in 1x PBS, followed by 2 hour-long incubation with corresponding secondary antibody diluted as 1:1000 [Alexa Fluor 555 (Thermo Fisher Scientific, no. A31570), Alexa Fluor 488 (Thermo Fisher Scientific, no. A21206), and Alexa Fluor 647 (Thermo Fisher Scientific, no. A32728)], (Thermo Fisher Scientific). After three washes in 1x PBS, the human fetal sections were permeabilized with 1x DEPC-PBS supplemented with 0.5% Triton X-100 for 10 min at RT. The section slides were then washed for 5 min in 1x DEPC-PBS, rehydrated for 10 min in 10% formamide supplemented with 2x SSC, and hybridized with specific probes (EIF4A2-cy3, SYP-cy3, SV2A-cy5, ST18-cy5, APBA1-cy3, NR4A2-cy3, NGFR-cy3, SYNPR-cy3) overnight at 37 °C using the Orbital Laboratory Shaker (Cleaver Scientific Ltd, no. CSL-NHYBRIDORB). The probes were diluted as 1:250 (final 4 ng/µl) in hybridization solution (50% formamide, 5X SSC, 5X Denhardt's solution, 500 ng/µl Salmon Sperm DNA, 250 ng/µl Yeast tRNA). The following day, the slides were washed two times in 10% formamide with 2x SSC for 30 min at 30 °C and then washed once in 1x DEPC-PBS for 5 min at RT. Lastly, autofluorescence quencher TrueBlack (Biotium no. 23007) was applied for 45 seconds on each slide, followed by one wash in 1X DEPC-PBS. Sections were covered with VECTASHIELD® Antifade Mounting Medium with DAPI (Vector Laboratories, no. H-1200-10). High-resolution scans of Nissl and adjacent immunofluorescence-labeled human sections were acquired with a Hamamatsu NanoZoomer 2.0 RS system using a 40x (NA 0.75) objective lens at 455 nm/pixel resolution. Fluorescence images were taken using the Hamamatsu LX2000 Lightning exciter, and processed using NDP.view2 Viewing software (U12388-01). Also, confocal imaging of immunoreactive cells in the subplate zone of the neocortex was performed using Olympus FV3000 microscope with 20x objective (UPlanSApo, NA 0.75, Olympus) and FV31S-SW Fluoview software at a resolution of 1024 ×1024 pixels.

### IF and FISH on mouse brain tissue
Mouse WT and *Emx1-Celf4* cKO P0 brains were fixed with RNAse-free 4% PFA for 8 h and cut coronally with a Leica VT1000S vibratome at 70 µm thickness under RNase free conditions. Selected sections were permeabilized with 500 µl of PBS-T (1X PBS with 0.5% Triton X-100) in sterile 24-well plates for 10 min at room temperature. Next, the sections were washed with 1X PBS, rehydrated in a 10% formamide solution supplemented with 2X SSC for 10 min at room temperature with 2 ng/µl of a DNA probe (cy5-90nt-cy5 or cy3-90nt-cy3, Integrated DNA Technologies). Using the HB-100 Hybridizer (UVP Laboratory Products), the sections were hybridized overnight at 37 °C in 300 µl of hybridization solution (50% formamide, 5X SSC, 5X Denhardt's solution, 500 ng/µl Salmon Sperm DNA, 250 ng/µl Yeast tRNA) and then washed three times for 20 min at 30 °C in 10% formamide supplemented with 2X SSC. Finally, the sections were immunostained with specific primary antibodies and corresponding secondary antibodies, as previously described[25].

### Immunohistochemistry (IHC)
IHC was performed using WT and *Emx1-Celf4* cKO coronal sections at E11, E13, E15, E17 and P0 as described previously in detail[25]. For fixation, embryonic brains at desired age were dissected and fixed in 4% PFA (pH 7.4; Sigma-Aldrich, no. 158127, St. Louis, MO, USA) in 1xPBS (Corning, no. 21-040-CV, Manassas, VA, USA) for 8 h at 4 °C. The brains were then washed three times in 1× PBS and preserved with 30% sucrose in 1xPBS for later use. For IHC, the brains were washed three times for 5 min in 1xPBS, embedded in 3% agarose (Lonza, no. 50004) and coronally sectioned at 70-80 µm using a Leica VT1000S vibratome.

Off-target antigens were blocked using the blocking solution [normal donkey serum (Jackson ImmunoResearch, no. 017-000-121), albumin (Biomatik, no. A2134), 0.2% Glycine (BDH, no. BDH4156), 0.2% L-lysine (Sigma, no. L5501), and 0.4%Triton (Sigma, X-100)] for 2–4 h at room temperature with gentle shaking. The tissue sections were incubated with a primary antibody solution diluted in blocking solution with 0.4% Triton X-100 for 16 hours at 4 °C with gentle rotation. Next, three washes for 5 min in 1x PBS were performed to remove primary antibody solution, followed by the incubation in the secondary antibody solution, which was also diluted in blocking solution, but without Triton X-100 for 2 hours with gentle shaking at room temperature. After three washes in 1x PBS, the sections were incubated for 10 min at room temperature with 1 µg/ml of DAPI (Thermo Fisher Scientific, no. D1306). Finally, DAPI solution was removed by performing two washes for 5 min in 1x PBS, and tissue sections were mounted with a coverslip using Vectashield mounting media (Vector Laboratories, no. H1000).

### CldU and IdU analogs administration for neurogenesis analysis
To generate timed pregnancies, pairs of females were housed with a single male overnight. In the morning of the next day, female mice were checked for vaginal plugs. The day of vaginal plug detection was termed E0.5. At E10.5 and E12.5, pregnant mice were injected intraperitoneally with freshly prepared chlorodeoxyuridine (CldU; Sigma; #C6891; dissolved in 1x PBS at 50 mg/kg), 24 h prior to sacrifice and embryonic brain isolation. The next day, freshly prepared iododeoxyuridine (IdU; Sigma; #I7125; dissolved in 7 mM NaOH in 1x PBS at 50 mg/kg) was injected 1 hour before extraction of pups and embryonic brain isolation. The brain sections were cut at 70 µm thickness to allow for CldU/ IdU antibody penetration. Immunostaining for Celf4 was performed first following our standard IHC protocol, as described in the relevant methods subsection. Next, antigen retrieval for thymidine analogs detection was performed to retrieve antigens masked by fixation. Sections were first treated with 1 M HCl for 15 min shaking at RT, followed by a 15 min stationary treatment with 2 M HCl. Acid was neutralized with 0.1 M Borax decahydrate (Sodium tetraborate; Sigma; #B-9876) twice for 10 min without shaking at RT. The sections were then washed with 1x PBS four times for 5 min each. Subsequently, our standard IHC protocol and cell quantification were performed according to the standard procedure described in the relevant methods subsection. Slides positive for CldU and IdU staining were imaged with a 20x objective.

### Confocal imaging
Mouse brain images were acquired with an Olympus BX61WI confocal microscope using either 20× objective to capture the entire cortical wall from pia to the ventricular zone, or 60x objective to capture the SP area and deep layer neurons. The Fluoview FV-1000 software was used for image processing. To allow for precise fluorescent intensity comparisons, all confocal images used in analysis or representative images were acquired with identical confocal settings per experiment. Brain images were merged and binned either in software Gimp2.10.14 or using FIJI distribution of ImageJ2 software (v2.3.0/1.53 f). Layer-marker positive cells and their migration profiles were determined by drawing a rectangle of standard width (200 pixels) that divides a cortical wall into ten equally sized bins from the top of the pia to the ventricular zone. Positively labeled neurons were normalized by the number of DAPI positive cell nuclei in the column.

### Primary and secondary antibodies
Primary antibodies and dilutions used on human fetal sections were: rabbit polyclonal anti-Celf4/ BRUNOL4 (dilution 1/250, Invitrogen, #PA5-58196, lot.no. UG2806225A); rat monoclonal anti-Ctip2 (clone: 25B6, dilution 1/500; Abcam; #ab18465, lot.no. #GR3272266-22); goat polyclonal anti-Nurr1/NGFI-Bβ/NR4A2 (dilution 1/250, R&D Systems, #AF2156, lot. no: UUW0318031); mouse monoclonal anti-Eif4a2 (clone:

G-5, dilution 1/400, Santa Cruz Biotechnology (SCB), #sc137147, lot.no: AO411); rabbit monoclonal anti-Sv2a (clone: D1L8S, dilution 1/400, Cell Signaling Technology, #66724 S, lot.no: 1); mouse monoclonal anti-TLE4 (clone: E-10, dilution 1/50; Santa Cruz; #sc365406, lot.no. #L1015); rabbit polyclonal anti-SERPIN1/Neuroserpin (dilution 1/200, Abcam, #ab330777); mouse monoclonal anti-Syp (clone: SY38, dilution 1/400, Invitrogen, #MA1-213, lot.no: WJ337774).

Secondary antibodies and dilutions used on human fetal sections were: Donkey anti-rabbit Alexa Fluor 488 (dilution 1/1000, Thermo-Fisher Scientific, #A-32790); Donkey anti-mouse Alexa Fluor 555 (dilution 1/1000, ThermoFisher Scientific, #A-32773); Goat anti-rat Alexa Fluor 555 (dilution 1/1000, ThermoFisher Scientific, #A-21434); Donkey anti-goat Alexa Fluor 488 (dilution 1/1000, ThermoFisher Scientific, #A-11055); Goat anti-mouse Alexa Fluor 488 (dilution 1/1000, ThermoFisher Scientific, #A-11001); Donkey anti-mouse Alexa Fluor 647 (dilution 1/1000, ThermoFisher Scientific, #A-31571).

Primary antibodies and dilutions used on mouse sections were: rabbit polyclonal anti-Celf4/ BRUNOL4 (IHC dilution 1/500, IF/FISH dilution 1/50, Invitrogen, #PA5-58196, lot.no. UG2806225A); mouse monoclonal anti-Eif4a2 (clone: G-5, dilution 1/400, Santa Cruz Biotechnology (SCB), #sc137147, lot.no: AO411); rabbit monoclonal anti-Sv2a (clone: D1L8S, dilution 1/300, Cell Signaling Technology, #66724 S, lot.no: 1); mouse monoclonal anti-Syp (clone: SY38, dilution 1/400, Invitrogen, #MA1-213, lot.no: WJ337774); mouse monoclonal anti-PSD95/ DLG4 (clone: K28/43, dilution 1/500, UC Davis/NIH NeuroMab Facility, #75-028, RRID:AB_2292909); rabbit polyclonal anti-vGlut1 (dilution 1/4000, Synaptic Systems, #135 302, lot. no: 1-41); guinea pig polyclonal anti-vGlut2 (dilution 1/2500, Synaptic Systems, #135 404, lot. no: 2-32); guinea pig polyclonal anti-VGAT (dilution 1/2000, Synaptic Systems, #131 004, lot. no: 2-43); mouse monoclonal anti-Gephyrin (dilution 1/100, Synaptic Systems, #147 021, lot. no: 1-26); rabbit polyclonal anti-Complexin3 (dilution 1/1000, Synaptic Systems, #122 302, lot. no: 1-8); goat polyclonal anti-Nurr1/NGFI-Bβ/NR4A2 (dilution 1/250, R&D Systems, #AF2156, lot. no: UUW0318031); rat monoclonal anti-Ctip2/Bcl11b (clone: 25B6, dilution 1/250, Abcam, #ab18465, lot. no: GR3272266-2); goat polyclonal anti-Brn1/POU3F3 (dilution 1/600, Novus Biologicals, #NBP1-49872, lot.no: P1 E210518); mouse monoclonal anti-Satb2 (clone: SATBA4B10, dilution 1/250, Abcam, #ab51502, lot. no: GR3174877-4); rabbit polyclonal anti-CDP/CUX1 (clone: M-222, dilution 1/250, Santa Cruz Biotechnology, #sc13024, lot. no: E0914); mouse monoclonal anti-Reelin (clone: G10, dilution 1/800, Millipore Sigma, #MAB5364, lot. no: 3099957); rat monoclonal anti-BrdU/CldU (clone: BU1/75 (ICR1), dilution 1/200, Abcam, #ab6326, lot. no: GR3365969-5); mouse anti-BrdU/IdU (clone: B44, dilution 1/100, BD Biosciences, #347580); goat polyclonal anti-Foxp2 (clone: N16, dilution 1/250, Santa Cruz Biotechnology, #sc-21069, lot. no: E0715); mouse monoclonal anti-Sv2/Sv2a (myeloma strain: SP2/0, dilution 1/1000, Developmental Studies Hybridoma Bank); mouse monoclonal anti-Gapdh (clone: 6C5, dilution 1/2000, Millipore Sigma, #MAB374, lot. no: 3189695).

Secondary antibodies and dilutions used on mouse sections were: Alexa Fluor® 488 AffiniPure Donkey Anti-Rabbit IgG (H + L) (dilution 1/250, Jackson ImmunoResearch, # 711-545-152); Cy™3 AffiniPure Donkey Anti-Rabbit IgG (H + L) (dilution 1/250, Jackson ImmunoResearch, #711-165-152); Cy™5 AffiniPure Donkey Anti-Rabbit IgG (H + L) (dilution 1/250, Jackson ImmunoResearch, # 711-175-152); Peroxidase AffiniPure Donkey Anti-Rabbit IgG (H + L) (dilution 1/1500, Jackson ImmunoResearch, #711-035-152); Alexa Fluor® 488 AffiniPure Donkey Anti-Mouse IgG (H + L) (dilution 1/250, Jackson ImmunoResearch, #715-545-150); Cy™3 AffiniPure Donkey Anti-Mouse IgG (H + L) (dilution 1/250, Jackson ImmunoResearch, #715-165-150); Cy™5 AffiniPure Donkey Anti-Mouse IgG (H + L) (dilution 1/250, Jackson ImmunoResearch, #715-175-151); Peroxidase AffiniPure Donkey Anti-Mouse IgG (H + L) (dilution 1/1500, Jackson ImmunoResearch, #715-035-150); Cy™5 AffiniPure Donkey Anti-Goat IgG (H + L) (dilution 1/250, Jackson ImmunoResearch, #705-175-147); Alexa Fluor® 488 AffiniPure Donkey Anti-Rat IgG (H + L) (dilution 1/250, Jackson ImmunoResearch, # 712-545-153); Cy™3 AffiniPure Donkey Anti-Rat IgG (H + L) (dilution 1/250, Jackson ImmunoResearch, # 712-165-153); Cy™3 AffiniPure Donkey Anti-Guinea Pig IgG (H + L) (dilution 1/250, Jackson ImmunoResearch, #706-165-148); Cy™5 AffiniPure Donkey Anti-Guinea Pig IgG (H + L) (dilution 1/250, Jackson ImmunoResearch, # 706-175-148).

### Cell quantification
For cell migration analysis and neurogenesis analysis, the neocortex was subdivided into 10-equally sized bins (bin 1 corresponds to marginal zone; bin 10 represents the ventricular zone) using software Gimp2.10.14. Binning conditions were kept constant across an experiment (the width of each bin was 200 μm) and the number of markers of interest+ cells in each bin was counted to be presented as the fraction of total DAPI+ cells or fraction of total glutamatergic identity marker of interest+ cells. Imaging and cell counting were done in double blind fashion where neither the person imaging nor quantifying knew the experimental condition.

### ThermoFisher CellInsight CX7 analysis
Image analysis was carried out on the CellInsight CX7 High-Content Screening (HCS) platform using the Neuronal Profiling analysis protocol. The Cx7 HCS parameters of VGAT/Gphn staining analysis are provided as an example in the online Supplementary data 12, as the same protocol parameters were applied in both vGlut1/PSD95 and vGlut2/PSD95 staining.

### Statistical analysis
Data normality was evaluated using Shapiro-Wilk test. For pairwise comparisons we used either parametric unpaired, two-tailed $t$ test with Welch's correction or non-parametric unpaired, two-tailed Mann–Whitney test. For multiple comparisons, we used one-way ANOVA, followed by post-hoc Brown-Forsythe (or Welch's ANOVA) or non-parametric alternative to one-way ANOVA, the Kruskal-Wallis test with Dunn's post-hoc test, both without matching or pairing. Statistical tests and the number (n) of replicates were noted in the figure legends. The significance threshold was set to $p < 0.05$ and is reported as: *$p < 0.05$; **$p < 0.01$; ***$p < 0.001$; ****$p < 0.0001$. The data were represented as graph bars with mean ± SEM. Statistical analysis was performed using the GraphPad Prism9 software (version 9.4.1).

### Reporting summary
Further information on research design is available in the Nature Portfolio Reporting Summary linked to this article.

## Data availability
The RNAseq and snRNAseq datasets generated in this study have been deposited at NCBI GEO, are publicly available as of the date of publication, and can be downloaded from the GEO database under accession number GSE214534. This SuperSeries is composed of the following SubSeries: GSE214272 (polysome RNAseq of human fetal neocortex), GSE214327 (human fetal neocortex CELF4 RIP-RNAseq), GSE214328 (polysome RNAseq of mouse neocortex; WTs and *Emx1-Celf4* cKOs), and GSE214532 (snRNAseq in human fetal neocortex). Human and mouse reference genomes were obtained from https://www.ncbi.nlm.nih.gov/grc. Reference dataset (ASD risk genes) used in this study is available as Supplementary Table 9 from[43]. Reference dataset (NDD risk genes set, DDG2P) is available from the Developmental Disorders Genotype-Phenotype Database (https://www.deciphergenomics.org/ddd/ddgenes). Functional annotations can be obtained from SynGO (https://syngoportal.org/). All data supporting the findings of this study are provided within the paper and its Supplementary Information. Source data are provided with this paper. Any additional information required to reanalyze the data reported in this

work is available from the lead contacts and Dr. Ronald Hart upon reasonable request. Source data are provided with this paper.

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

## Acknowledgements

This work was supported by National Institutes of Health (NIH) grants (NS064303 and NS075367) and Robert Wood Johnson Medical School start-up funds (M.R.R.). This work was also supported in part by the Assistant Secretary of Defense for Health Affairs endorsed by the Department of Defense, through the Epilepsy Research Program under Award No. W81XQH-18-1-0338 to M.R.R, as well as in part by the NIH Grant U10AA008401 from the National Institute on Alcohol Abuse and Alcoholism (NIAAA) to RP.H. Opinions, interpretations, conclusions and recommendations are those of the author and are not necessarily endorsed by the Department of Defense. The U.S. Army Medical Research Acquisition Activity, 820 Chandler Street, Fort Detrick MD 21702-5014 is the awarding and administering acquisition office. In addition, this work was supported by the "Research Cooperability" Program of the Croatian Science Foundation funded by the European Union from the European Social Fund under the Operational Programme Efficient Human Resources 2014-2020 (PSZ-2019-02-4710 to Ž.K. and M.R.R.) and co-financed by the Scientific Center of Excellence for Basic, Clinical and Translational Neuroscience project "Experimental and clinical research of hypoxic-ischemic damage in perinatal and adult brain" (GA KK01.1.1.01.0007) funded by the European Union through the European Regional Development Fund (M.J). I.S. was supported by The Governor's Council for Medical Research and Treatment of Autism's FY22 Autism funding opportunity (CAUT22AFP010). S.D.R. is supported by the Beatrice & Samuel A Seaver Foundation, M.G.-F. is supported by the La Fundación Alfonso Martín Escudero. The authors acknowledge the Office of Advanced Research Computing (OARC) at Rutgers, The State University of New Jersey for providing access to the Amarel cluster and associated research computing resources. We thank Donna Tran, Sohan Saha, Kacey Balseca and Omar Zahara for technical support and Dr. Paul Copeland (Rutgers University) for microultracentrifuge usage.

## Author contributions

I.S., Y.P., Ž.K. and M.R.R. conceived the study. I.S. designed and performed the experiments (snRNAseq and polysome RNAseq experiments and pre-processing, IHC, immuno-EM, SNs isolation, WB, RT-PCR, designed the mutant Emx1-Celf4 line), analyzed and interpreted the data, provided trainee supervision, made publication figures, wrote the original draft, edited, revised and proofread the manuscript. Y.P. designed and performed the experiments (RIPs, IHC, FISH, RT-PCR), analyzed and interpreted the data, contributed to writing the discussion, and edited the original manuscript. T.M. carried out human RIP and human immuno-EM experiments. J.K. performed IHC and IF on human tissue and confocal imaging. P.M. and J.H.M. performed CX7 analyses. N.F.P. and A.R. contributed to colony maintenance, data quantification, provided edits to the manuscript. G.W.M. provided training in immuno-EM. J.F. and P.S. analyzed mouse RIP-RNAseq and polysome RNA-seq data. M.G.-F. provided training in SNs isolation. M.J. provided financial support. S.D.R. performed enrichment analyses. I.K. and S.D.R. provided edits and comments to the manuscript. R.P.H. analyzed human snRNA-seq and mouse/human polysome RNAseq data, and provided direction, edits, and comments to the manuscript. Ž.K. and M.R.R. provided oversight, edits, and comments to the manuscript. All authors had the opportunity to comment on the manuscript.

## Competing interests

P.S. is a Director at Ananke Therapeutics, a Scientific Advisory Board member of Trestle Biosciences, and consults for Ribo-Therapeutics. Other authors declare no competing interests.
