## [Peer Review File · Nature Communications]

Celf4 controls mRNA translation underlying synaptic development in the prenatal mammalian neocortex.REVIEWER COMMENTS

Reviewer #1 (Remarks to the Author):

In this manuscript, Salamon I et al., study translational control in prenatal cortical development. The authors combined snRNA-seq and polysome profiling of developing brains to identify translationally controlled transcripts during development. The authors then focused on CELF4 and showed CELF4 acts as a translation repressor of many synaptogenesis genes. They used RIP-seq to compare human and mouse CELF4 targets and found significance difference. Finally, the authors showed Celf4 conditional KOs alter synapse density in a sex-specific manner. Overall, this is an interesting paper with rich datasets. It will be stronger if the genomics and/or molecular analysis can somehow explain the Celf4 KO phenotype. Below are some comments to tighten up the analyses.

Major:

To define translational regulation the authors selected developmentally unchanged mRNAs to analyze their polysome association. This approach can have many false negatives because translation regulation may be concurrent to transcriptional control, and have many false positives when transcription regulation is at subthreshold and considered unchanged but contributes to the signals of polysome association. A more rigorous approach would be normalizing polysome association with free mRNA abundance before considering developmental regulation.

The authors may need to be cautious in claiming cell type specific translational regulation based on scRNA-seq. While scRNA-seq is robust to transcriptomically define cell clusters, they are not robust to discern gene expression differences of each gene, or the lack thereof, between clusters.

Unlike CLIP-Seq, RIP-seq does not necessarily identify targets directly bound by CELF4. The authors need to be careful in their interpretation.

Related to the above comments, most seq experiments have n=2. It is understandable human samples are difficult to obtain. What the authors can do to minimize artifacts of biological variability is to increase thresholds for the analysis.

Abstract: "... in agreement with unique features of human NDDs". Maybe I missed this, what unique features are explained by the human-specific celf4-targets?

"CELF4 can serve as a novel and reliable marker of human SP and deep layers throughout neocortical development." this needs evidence, eg, costaining with SP molecular marker in fig2b.

Fig2c: can the authors quantify % SP neurons expressing CELF4 to support their conclusion.

Fig4b-e: overlapping staining needs to be quantified, particularly when comparing wt with KO in fig4e.

The selective increase of SV protein in synaptoneurosomes but not in whole cortex lysates is interesting, because CELF4 is also highly expressed in the cytoplasm. What can be the possible explanation?

The sex-specific phenotype is interesting but seems a bit disconnected from the rest of the manuscript, particularly when these are not Celf4 targets. Is the phenotype related to the celf4-dependent translation control?

Reviewer #2 (Remarks to the Author):

The Salamon et al., paper presents a comprehensive single-nucleus RNA sequencing (snRNAseq) map across early fetal and midfetal human neocortical development. The authors explored the development of frontal neocortical cell populations, including the SP neurons.

The study provides valuable polysome profiling coupled to snRNAseq of human prospective prefrontal neocortices at three developmental periods; from early fetal [11/12 post-conceptual weeks (PCW)], early midfetal (14/15 PCW) to late midfetal (17/18 PCW). Human mRNAs undergoing translational control in distinct cell populations of developing prefrontal neocortices. These data suggest that human neocortical development is heavily regulated by complex mRNA translation mechanisms that are stage-specific.

The study homes in to the CELF4, encoded by a risk gene for ASD and related NDDs. This gene is expressed in the developing subplate. CUGBP Elav-Like Family Member 4 (CELF4), is expressed in neurons of two compartments which are enriched in the initial synaptogenesis, including the evolutionarily advanced neocortical compartment, the subplate (SP).

The study explores the differences between mouse and human Celf4/ CELF4 synaptic mRNA targets. The forebrain-specific Celf4 deletion from mouse cortical progenitors and postmitotic neurons disrupts the balance of SP synapses. Moreover, the study shows that this disruption is sex-specific.

The study further reinforces the notion that has been published in previous studies that ASD risk genes are enriched in maturing and mature excitatory and inhibitory neurons, as well as in the progenitor clusters in the presented snRNAseq dataset.

SynGO revealed that 22 (out of 129) genes from the late SP clusters and 20 (out of 125) genes expressed in the late excitatory layer 5/6 clusters were uniquely mapped to SynGO annotated genes, as compared to “brain” selected background.

Overall, the paper contains extremely valuable and interesting data. Huge amount of work went into the study. It has enough data to produce 2-3 high impact papers. Having said all this, I would like to point out some shortcomings of the presentation. This can be easily improved with revisions.

1. The title is rather general. It would be helpful to give a more specific title that covers the actual findings more closely.

2. The introduction also covers a huge area, and it could be more focussed on the key elements of the actual study. It is almost as the authors published 3 papers in one publication; the transcriptomic atlas that eventually explores the ASD susceptibility genes and a separate study when the actual function of a single gene is studied in mouse and then the sex specificity of this gene action is described, but not explored in detail. The authors could have written 3 papers on these respective topics, specifically aiming to well distinguished area. Current manuscript would like to present all this, and we end up with a rather complicated story that is very difficult to follow. I struggled reading the paper, although I am highly interested in this area of research. Just the comparison of the cellular transcriptomes with translomes obtained from prenatal human prefrontal cortex would deserve publication in detail.

Analysis of sex specificity of these transcriptomes and translomes could have been done on the entire database, and not just on the consequences of the ablation of a single gene.

Overall, this article is far too long and complex to be presented as one single study. This work can be split into different publications, and in doing so the authors might stress more onto the quality and coherency of the data they present. Furthermore, the text is cumbersome, and the reading is particularly heavy. This could be improved with accurate proofreading and editing before submission. Some examples for the text that will need attention: in line 160-161 the term “corroborate” is repeated. In line 272 “coincides in concomitant” is another repetition. The technique used for candidate validation presented in paragraph #4 is described in a confusing manner. Some comments are inappropriately located, as they have been added to the text a posteriori and without paying attention as to whether they sound timely or not (i.e. introducing a result paragraph with a generalized conclusion like in lines 174-175 “Comparing the levels of free mRNAs (input) with those bound to polysomes revealed that translational regulation couples with transcription to define human neocortical development” sounds like jumping to a conclusion too fast and before actually presenting the data). Similarly, in line 412 “Data suggest that Celf4 expressing neurons are some of the earliest born neurons and serve an essential role in neurogenesis” the statement is incorrect, as the data simply clarify the neurons’ birthdating, without providing any relevant information on their role in neurogenesis as their function has not been experimentally manipulated -which instead is what presented in the following paragraph.

The content of the paper could be organised in a much more systematic manner: jumping from human datasets to a comparison with mouse data makes it difficult to grasp the final message. Overall, the general organization and excessive length of the text tends to prevail, and this overshadows the genuine originality and strength of the excellent data.

3. The rest of the study explores the involvement of Celf4 in synaptic function and suggest that Celf4 might act as a translational repressor of synaptic mRNAs. This part of the study would be sufficient for a high-quality paper. It clearly demonstrates the importance of Celf4 availability for translation initiation and suggest that Celf4 deletion causes specific changes in translational control by favouring the shift of synapse-related mRNAs into the polysome-bound fractions.

4. The sex differences are only analysed for a single gene. The authors found that non-nuclear VGAT+Gphn+ overlapping area was significantly increased in male mutants, while the opposite effect was observed in female mutants. Most of the description of the phenotype is at P0. It would be interesting to explore these differences at additional ages and extend the study to more genes.

5. On several occasions, the authors fail to explain the reasons behind some of their choices and/or selections of data for specific analysis. For instance, in the first paragraph, it is not clear why they did not check for the risk-gene set enrichment with an unbiased analysis across the entire snRNAseq dataset (i.e. all the clusters) instead of sub-setting the SP. Although it might be a relevant and understandable choice, without an explanation it gives the impression of a biased selection, especially considering that the authors themselves comment on the fact that the SP does not have enrichment for risk genes

associated with severe forms of NDD.

In the same section, the authors comment [lines 176-177] that a change in translation state might be driven by a change in cell-type composition, however, they do not clarify why they chose to do a bulk RNAseq considering that it might not be the best experimental choice in such instance. They also explore the intersection of snRNAseq and polysome-associated changing transcripts at different stages, without specifying why they did not perform the analysis also with the monomer-associated transcripts. In the same paragraph, they do not explain why they chose to do the GO enrichment analysis only in the late cell clusters, instead of doing it also for the early ones.

In section #7 the authors decide to have a closer look at upper-layer cellular subtype markers, without explaining the relevance of such cells at all – the entire focus of the paper is in subplate/deep-layer neurons considering that the protein of interest is expressed there, so it would have made more sense to look at those compartments more closely instead.

These are just few examples from the results, but there are many of these instances in the text, which brings it to the same conclusion: splitting the paper into different works might be helpful in increasing accuracy and efficiently to convey the compelling results described by the authors.

6. In some instance, the results do not support the claims made by the authors. For example, in paragraph #2 the authors conclude that a translational control mechanism might be responsible for the more rapid maturation of the human SP vs upper and lower neocortical layers potentially by redefining its synaptic protein repertoire from 1st to 2nd trimester. This contrasts with the choice of analysing only the late SP clusters, rather than looking also at the earlier stages. In paragraph #7, the finding that the production, migration, and laminar organization of cortical neurons is not affected is in direct contrast with the previous comment regarding an “essential” role played by the molecule. In section #3 the authors search for candidate molecules expressed in both SP and L5/6, disregarding what they previously presented about the central role of the SP itself. It doesn't make any sense to group together SP and L5/6, unless further explained and justified. The authors write: “CEL4 can serve as novel and reliable marker of human SP and deep layers throughout neocortical development.” If CEL4 is expressed in several cell populations than what is it marker of?

The Salamon et al., manuscript contains highly interesting and valuable data and I am supportive in its publication. Nevertheless, I have to say that the paper contains too much data that are not all necessary to present together. The manuscript is hard to follow, and it is not an easy read. I urge the authors to streamline the story and consider presenting the data in separate paper. They could consider 2-3 excellent paper with clear and concise messages.

Reviewer #3 (Remarks to the Author):

The authors report polysome profiling (mRNA analysis that are associated with polysome) together with single nuclear RNAseq of human developing prefrontal neocortex and identified a set of human mRNAs undergoing active translation in distinct cell populations, which include mRNAs encoding synaptic proteins. Among those highly expressed in the subplate (SP) neurons, they found that an RNA binding protein, CELF4, targets a group of synaptic genes that are known risk molecules for adverse neurodevelopmental outcomes, such as autism spectrum disorder. By examining both murine and human neocortex by RIP-RNAseq, by which one can detect particular protein-bound (targeted) mRNA, the authors uncovered common or distinct CELF4 targets between these two species. The unexpected last highlight of this paper is that they found the forebrain-specific *Celf4* deletion from mouse cortical progenitors and postmitotic neurons disrupts the abundance of both excitatory and inhibitory synapses in a sex-specific fashion. Although findings are well described and provide high impact to the science community, I find a premature analytic approach in snRNAseq data. Contextually it will not change the conclusion of the paper, however, I would request the authors to re-examine the snRNAseq data in a more sophisticated manner. For example,

(1: Fig.1a, Supplementary Fig.1) All six dataset should be integrated and processed for the downstream filtering and clustering analysis (Seurat provides such a framework).

(2: Fig.1a and line 136-) To objectively assess developmental changes in snRNA clusters, the authors need to perform pseudotime analysis e.g. using Monocle or Slingshot.

(3: Fig.2a) The discovery process of CELF4 should be more objective. One approach would be to perform a differential expression analysis between SP and other neuronal clusters. The authors should provide statistics for unbiased evaluation of the SP-specific gene(s) (Seurat provides such a framework).

(4: Fig.2h) Co-expression of CELF4 and its candidate target mRNAs at single cell level should be examined more objectively (by using e.g. COTAN

<https://academic.oup.com/nargab/article/3/3/lqab072/6348150>). The representation in the provided figure is not convincing enough to draw conclusions.

Other minor concerns are as follows,

(5: Fig.1f) I find a conceptual contradiction in intersecting snRNAseq data and polysomal RNAseq data, since they are derived from different subcellular origins; the former is from the nucleus and the latter is from cytosol. Previous reports support that the RNA levels are mostly concordant between nucleus and cytosol (e.g. <https://www.nature.com/articles/s41598-017-04426-w>) and I would add such a paragraph mentioning the authors performed this analysis based on the assumption that the RNA levels are concordant between nucleus and cytosol.

(6) lines 125-126, “bulk transcriptomic data” should be corrected to “single nuclei transcriptomic data” or something.

(7) lines 174-175, “Comparing the levels of free mRNAs (input) with those bound to polysomes revealed that translational regulation couples with transcription to define human neocortical development” This statement sounds too definitive. While this finding suggests the importance of translation control in human neocortical development, it may be misleading to say that it defines human neocortical

development.

(8) line 200, “expression cutoff frequency to 30% of cells within a cluster” should be missing the cutoff criteria for log₂FC increase threshold (default setting by Seurat is 0.25 but the method describes “1.2-fold cutoff” so please clarify.

(9) There are some typos or sentences that are a bit hard to interpret. Proofreading by a trained yet unbiased staff member is recommended.

(10) I would request the authors to describe more on the phenotype of the Celf4-cKO mouse. I would discuss how the “recurrent seizures later on” (line 698) is comparable to the previous findings on Celf4 KO mouse (e.g. <https://www.sciencedirect.com/science/article/abs/pii/S0006291X22003989>) and relate it to human neurodevelopmental disorders. I would also request to mention whether there is a sex difference in the occurrence of seizures in Celf4-cKO mouse.

We are immensely grateful to Reviewers for the encouraging opinions on our first submission. We appreciate the effort and time you have committed to providing us with detailed and constructive feedback that was tremendously helpful in strengthening our paper. We have marked all the changes made in the manuscript using the track changes feature in red, and uploaded figures to reflect all the suggestions provided by the Reviewers. Here is a point-by-point response to your suggestions:

Reviewer 1 (Remarks to the Author):

Reviewer 1 Comments:

(1) To define translational regulation the authors selected developmentally unchanged mRNAs to analyze their polysome association. This approach can have many false negatives because translation regulation may be concurrent to transcriptional control, and have many false positives when transcription regulation is at subthreshold and considered unchanged but contributes to the signals of polysome association. A more rigorous approach would be normalizing polysome association with free mRNA abundance before considering developmental regulation.

Authors response: Thank you for your comments and concerns. We agree that choosing the appropriate approach to define transcripts that are only under translational control is of great importance. Indeed, our analysis might be partially affected by some false negatives (translational activity is regulated independently of the free mRNA levels) and some false positives (mRNAs at the subthreshold levels that are transcriptionally changed but included in the analysis) identified by the Reviewer. We believe that by filtering out those transcripts with major changes at the transcript level (either due to changes in the transcription itself or mRNA stability), we were best able to identify relatively “unchanged” mRNAs that are hypothesized to be subject to translational control. As the Reviewer suggested, we attempted to normalize polysome-associated transcripts with free mRNA abundance; however, due to the resulting increased variability, no genes could be declared as significantly different (please see image below). Therefore, we choose to retain our original logic but we have expanded the Discussion to explain our strategy and consider these concerns as follows:

“Standard RNAseq based on polysome fractionation analysis is an effective and well-established method to monitor the translational status of mRNAs but is also recognized to have some limitations. Arguably, choice of normalization method by which experimental variations are corrected can have significant impact on the downstream analysis results. For example, our attempt to normalize polysome-associated transcripts with free mRNA abundance resulted in the increased signal-to-noise ratio, masking any genes that could be declared as significantly different. In contrast, filtering out those transcripts with major changes at the transcript level (either due to changes in the transcription itself or mRNA stability) enabled identification of relatively unchanged mRNAs whose expression may be regulated by the translational control mechanisms.”

(2) The authors may need to be cautious in claiming cell type specific translational regulation based on scRNA-seq. While scRNA-seq is robust to transcriptomically define cell clusters, they are not robust to discern gene expression differences of each gene, or the lack thereof, between clusters.

Authors response: Thank you for bringing up this important point. We understand that this is somewhat controversial in the field even though snRNAseq data are frequently used to evaluate differential expression. Our primary interpretation is to compare differences between developmental phases within a group of clusters, with a secondary interpretation parsing differences in polysome profiling results (from bulk RNAseq) to selected cell clusters. As the Reviewer suggests, we added language to the Discussion to point out the shortcomings of this approach:

“The analysis of snRNAseq data also presents some challenges, especially when estimating differential expression of each gene, or the lack thereof, at often sparse and noisy single nucleus resolution. While this limitation needs to be improved, snRNAseq technology is still the most common approach to analyze the transcript expression of single cells.”

(3) Unlike CLIP-Seq, RIP-seq does not necessarily identify targets directly bound by CELF4. The authors need to be careful in their interpretation. Related to the above comments, most seq experiments have n=2. It is understandable human samples are difficult to obtain. What the authors can do to minimize artifacts of biological variability is to increase thresholds for the analysis.

Authors response: We thank the Reviewer for highlighting this very valid point. We performed RIP-RNAseq instead of CLIP-RNAseq because we wanted to purify CELF4-mRNA complexes under the native conditions. While we used three distinct human neocortices for most of RIP pulldown experiments (i.e. early fetal and late midfetal phases), early midfetal phase indeed had smaller sample size (n = 2) due to the limitation of available material. While a previous study using human embryonic stem cells reported RIP RNA-seq cutoff of 1.5-fold (Byres et al., 2021), in the revised

version of our manuscript we followed the Reviewer's suggestion and increased the threshold for the analysis from > 2 fold-change to > 3.249 fold-change to make analyses more stringent. We considered the 227 early-, 1536 mid- and 763 late-target mRNAs identified using the revised approach (> 3.249-fold change cutoff or 1.7x in log₂ scale) as *bona fide* CELF4 target mRNAs, especially as some of them were validated by RIP qPCRs (as shown in the Figure 2g). As pointed by the Reviewer, we recognize that the limitation of the RIP-seq compared to CLIP-seq should be mentioned in the paper, so we updated the following sentence in the Results section:

“To identify human mRNA targets of RBP CELF4 during neocortical development, we conducted native RNA immunoprecipitation (RIP) on early fetal (11/12 PCW), early midfetal (14/15 PCW), and late midfetal (17/18/20 PCW) neocortices using the validated CELF4 antibody or corresponding negative control immunoglobulin G (IgG). RIP samples were then subjected to RNAseq (GEO accession GSE214327). Using stringent criteria [\log_2FC (CELF4/IgG RIP) > 1.7 and adjusted p-value < 0.05], we observed a total of 227 target mRNAs in the early phase, while a much higher number of putative CELF4 targets was detected in the mid (a total of 1536 mRNAs) and late (a total of 763 mRNAs) phases (Fig. 2e and Table S8). Among these mRNAs, 45 early, 183 mid and 125 late CELF4 candidates encode presynaptic and postsynaptic proteins, based on SynGO annotations⁶⁸ (Table S8b-d).”

The results of new Gene Ontology enrichment analyses are presented in the revised Figures 2e and 2f (hCELF4 RIP-RNAseq Chord diagrams). In Supplementary table 8a, we also added the list of read counts mapped to each gene for each sample (CELF4 and IgG pulldowns). Results of SynGO analyses using revised CELF4 targets were presented in Supplementary table 8b-e. Lastly, we updated the intersection analyses between mouse *Celf4* targets and phase-specific hCELF4 targets above the 3.249-fold change cutoff. The results of these analyses yielded similar results as previously described. The revised Venn diagrams and Chord diagrams are provided in Figure 3c. We also added Supplementary table 8e that contains the list of shared *Celf4*/CELF4 targets. The sentences in the main text and figure captions were updated accordingly to reflect all revised changes.

(4) Abstract: “... in agreement with unique features of human NDDs”. Maybe I missed this, what unique features are explained by the human-specific *celf4*-targets?

Authors response: Thank you for this important comment. We apologize for the oversight and have removed the following sentence from the abstract: “Importantly, we found evolutionary differences between mouse and human *Celf4*/ CELF4 synaptic mRNA targets, in agreement with unique features of human NDDs.”

(5) “CELF4 can serve as a novel and reliable marker of human SP and deep layers throughout neocortical development.” this needs evidence, eg, costaining with SP molecular marker in fig2b.

Authors response: Thank you for your observation. We agree it is important to provide this evidence, therefore we have updated Figure 2b to include results showing the co-localization of CELF4 with the deep layer and SP marker CTIP2. To additionally bolster our conclusion that CELF4 can serve as a novel marker of the deep layers and SP area, we added new figures (Supplementary figures 4a and 4b) corroborating the co-localization of CELF4 with the established markers for deep layers and the SP – TLE4 and neuroserpin. The main text and figure captions are updated accordingly. We hope that these new findings have strengthened our statement.

(6) Fig2c: can the authors quantify % SP neurons expressing CELF4 to support their conclusion.

Authors response: We thank the Reviewer for this suggestion. Unfortunately, the descriptive results presented in Figure 2c cannot be used reliably for quantitative analyses. Although we did observe almost complete co-localization between CELF4 and *ST18* in SP neurons, the small sample size (n = 2 neocortices) makes it difficult to draw any strong conclusions about the percentage of their co-expression. While we agree with the Reviewer's suggestion and we understand that the small sample size may be a limiting factor in this study, human fetal cortical tissue is very precious and hard to obtain – the main reasons why we cannot perform additional experiments. Instead, we decided to quantify the percentage of Celf4+Ctip2+ cells among the total labeled Celf4+ cells in developing neocortex at E15.5 and E17.5 developmental stages in mouse embryos. We have included the new graphs as Supplementary figure 5b and updated the main text and figure captions accordingly:

Main text: “At both E15.5 and E17.5, almost all of the Celf4+ cells were localized in the SP area (Supplementary fig. 5b, left and middle) as indicated by the near-complete co-localization of Celf4 and Ctip2 staining (Supplementary fig. 5b, right).”

Figure captions: “(b) **Left and middle:** Quantification of the distribution of Celf4+ (dark green) and Celf4+Ctip2+ (light green) in E15.5 and E17.5 neocortex determined as a percentage of total number of Celf4+ cells. The grid was divided into 10 bins of equal height from pia (above MZ) (bin 1) to ventricular zone (bin 10). Sample size: n = 3 animals per developmental stage. **Right:** Quantification of the proportion of Celf4+Ctip2+ cells from total labeled Celf4+ cells in developing neocortex at E15.5 and E17.5.”

(7) Fig4b-e: overlapping staining needs to be quantified, particularly when comparing wt with KO in fig4e.

Authors response: We appreciate the Reviewer's suggestion. We agree that figures should be quantified, and we performed the analysis whenever possible. We successfully quantified the percentage of Eif4a2_{weak}+Sv2a_{strong}+ SP neurons or Eif4a2_{strong}+Sv2a_{weak}+ SP neurons over total number of Eif4a2 neurons in Figure 4e and the corresponding quantitative graphs are included as a part of Figure 4e. The main text and figure captions are also adjusted accordingly. Figures 4b-d represent the qualitative observations and cannot be used for statistical analysis due to the small sample size (n = 2 neocortices) (please see the comment above).

(8) The selective increase of SV protein in synaptoneurosomes but not in whole cortex lysates is interesting, because CELF4 is also highly expressed in the cytoplasm. What can be the possible explanation?

Authors response: We thank the Reviewer for bringing this intriguing observation to our attention. This might be the result of selective protein transport to the synapses, where Sv2a proteins are most needed, or Celf4-mediated translational regulation locally at synapses. In line with the second possibility, Celf4 protein is present not only in neuronal somata with a strong cytoplasmic pattern but also in axons or dendrites of developing glutamatergic neurons (Fig. 3a, Supplementary Fig. 5 and Supplementary Fig. 8). A previous study also suggested that *Celf4* deletion caused the differences in the relative protein abundance of some of its targets between the somatic cytoplasm and neurites in the developing neocortex (Wagnon et al., 2012). Other RNA-binding proteins, such as FMRP (DICTENBERG et al., 2008; FEUGE et al., 2019), ZBP1 (ZHANG et al., 2001; PERYCZ et al., 2011), Pumilo2 (MARTÍNEZ et al., 2019) and HuD (TIRUCHINAPALLI et al., 2008) have also been reported to have similar roles in mRNA trafficking and local translation in the neurites. Further research would be necessary to answer these questions, which is our future direction.

(9) The sex-specific phenotype is interesting but seems a bit disconnected from the rest of the manuscript, particularly when these are not Celf4 targets. Is the phenotype related to the celf4-dependent translation control?

Authors response: We thank the Reviewer for this comment. We believe that identifying the mechanisms of the sex differences we report in Figure 5 is beyond the scope of the current manuscript, given the extent of the putative target mRNAs regulated by Celf4/CELF4 and the potential secondary consequences on other downstream mRNAs. Figure 5 serves as a confirmation that Celf4 has a significant effect on the synapses, and that early-onset developmental perturbations could further cause phenotypes that appear later at postnatal stage. We also believe that it was important to report these observations given the prominent sex differences in prevalence reported in ASD. This intriguing question is indeed subject of future studies.

References:

- Byres, L. P., Muftuev, M., Yuki, K. E., Wei, W., Piekna, A., Wilson, M. D., et al. (2021). Identification of TIA1 mRNA targets during human neuronal development. *Mol Biol Rep* 48, 6349–6361. doi: 10.1007/s11033-021-06634-0.
- Dictenberg, J. B., Swanger, S. A., Antar, L. N., Singer, R. H., and Bassell, G. J. (2008). A Direct Role for FMRP in Activity-Dependent Dendritic mRNA Transport Links Filopodial-Spine Morphogenesis to Fragile X Syndrome. *Developmental Cell* 14, 926–939. doi: 10.1016/j.devcel.2008.04.003.
- Feuge, J., Scharkowski, F., Michaelsen-Preusse, K., and Korte, M. (2019). FMRP Modulates Activity-Dependent Spine Plasticity by Binding Cofilin1 mRNA and Regulating Localization and Local Translation. *Cerebral Cortex* 29, 5204–5216. doi: 10.1093/cercor/bhz059.
- Martínez, J. C., Randolph, L. K., Iascone, D. M., Pernice, H. F., Polleux, F., and Hengst, U. (2019). Pum2 Shapes the Transcriptome in Developing Axons through Retention of Target mRNAs in the Cell Body. *Neuron* 104, 931-946.e5. doi: 10.1016/j.neuron.2019.08.035.
- Perycz, M., Urbanska, A. S., Krawczyk, P. S., Parobczak, K., and Jaworski, J. (2011). Zipcode Binding Protein 1 Regulates the Development of Dendritic Arbors in Hippocampal Neurons. *J. Neurosci.* 31, 5271–5285. doi: 10.1523/JNEUROSCI.2387-10.2011.
- Tiruchinapalli, D. M., Ehlers, M. D., and Keene, J. D. (2008). Activity-dependent expression of RNA binding protein HuD and its association with mRNAs in neurons. *RNA Biology* 5, 157–168. doi: 10.4161/rna.5.3.6782.
- Wagon, J. L., Briese, M., Sun, W., Mahaffey, C. L., Curk, T., Rot, G., et al. (2012). CELF4 regulates translation and local abundance of a vast set of mRNAs, including genes associated with regulation of synaptic function. *PLoS Genet* 8, e1003067. doi: 10.1371/journal.pgen.1003067.
- Zhang, H. L., Eom, T., Oleynikov, Y., Shenoy, S. M., Liebelt, D. A., Dictenberg, J. B., et al. (2001). Neurotrophin-Induced Transport of a β -Actin mRNP Complex Increases β -Actin Levels and Stimulates Growth Cone Motility. *Neuron* 31, 261–275. doi: 10.1016/S0896-6273(01)00357-9.0

Reviewer 2 (Remarks to the Author):

Reviewer 2 Comments:

(1) The title is rather general. It would be helpful to give a more specific title that covers the actual findings more closely.

Authors response: Thank you for pointing this excellent suggestion. We changed the title of our manuscript to: “Celf4 controls mRNA translation underlying synaptic development in the prenatal mammalian neocortex.”

(2) The introduction also covers a huge area, and it could be more focussed on the key elements of the actual study. It is almost as the authors published 3 papers in one publication; the transcriptomic atlas that eventually explores the ASD susceptibility genes and a separate study when the actual function of a single gene is studied in mouse and then the sex specificity of this gene action is described, but not explored in detail.

The authors could have written 3 papers on these respective topics, specifically aiming to well distinguished area. Current manuscript would like to present all this, and we end up with a rather complicated story that is very difficult to follow. I struggled reading the paper, although I am highly interested in this area of research. Just the comparison of the cellular transcriptomes with translomes obtained from prenatal human prefrontal cortex would deserve publication in detail. Analysis of sex specificity of these transcriptomes and translomes could have been done on the entire database, and not just on the consequences of the ablation of a single gene. Overall, this article is far too long and complex to be presented as one single study. This work can be split into different publications, and in doing so the authors might stress more onto the quality and coherency of the data they present. Furthermore, the text is cumbersome, and the reading is particularly heavy. This could be improved with accurate proofreading and editing before submission.

Authors response. We appreciate that Reviewer feels that this study represents a large set of findings that could be potentially represented as three manuscripts. We do regret that the text was not clear, and we have prepared a revised version that we hope is clearer and more easily understandable. In addition, the 1st co-author graduated and is moving to her new postdoctoral position at Yale University. Although it would be beneficial, extending the presented results into three manuscripts is unfortunately not feasible.

Some examples for the text that will need attention:

→ in line 160-161 the term “corroborate” is repeated.

Authors response: Thank you for pointing this out. We edited this paragraphs with the following sentence in the manuscript:

“These data support previous evidence that the midfetal cortical development is a nexus of high vulnerability for ASD and NDDs^{41,42,45-49} and further implicate SP neurons in the etiology of ASD.”

→ in line 272 “coincides in concomitant” is another repetition.

Authors response: Agree. We edited the sentence as follows:

“Additionally, the CELF4 protein expression coincides with the expression of established SP enriched markers – NR4A2, ST18 and SERPIN1^{20,71}”.

→ the technique used for candidate validation presented in paragraph #4 is described in a confusing manner.

Authors response: Thank you for this comment. To clarify candidate validation to our future readers, we revised the paragraph, as follows:

“To validate some of these putative CELF4 mRNA targets, we performed quantitative real-time PCR (qRT-PCR) using RNAs of RIP-RNaseq experiments. Sequence-specific primers (Table S11) were designed for several CELF4 target candidates encoding presynaptic (SYNPR, SYN2, SV2A, SYP, APBA1) and postsynaptic (GABRA3) proteins, transcription factors (TLE4, NR4A2, ST18), or translation initiation factors (EIF4A2) in the SP area. Enrichment of selected mRNA targets by RIP-qRT was calculated as the fold-change bound by CELF4 from the non-specific binding in IgG RIP. Since the remaining quantities of purified RNAs from each developmental-phase specific CELF4 RIPs were limited after RNaseq, we grouped together the results of all developmental phases obtained from each RIP-qRT experiment. As expected, the expression levels of internal control ACTB and negative control NES were not found enriched in CELF4 RIP samples, confirming the specificity of the approach. In contrast, selected CELF4 mRNA candidates (SYNPR, SYN2, SV2A, SYP, APBA1, GABRA3, TLE4, NR4A2, ST18 and EIF4A2) were significantly enriched in CELF4 RIPs (Fig. 2g). Some of these mRNAs were monitored for expression in the developing fetal neocortex and showed co-localization with CELF4 protein in the SP area (Supplementary Fig. 4c). These results confirm that CELF4 binds a set of synapse-associated mRNA targets in vivo in human developing neocortices.”

→ some comments are inappropriately located, as they have been added to the text a posteriori and without paying attention as to whether they sound timely or not (i.e. introducing a result paragraph with a generalized conclusion like in lines 174-175 “*Comparing the levels of free mRNAs (input) with those bound to polysomes revealed that translational regulation couples with transcription to define human neocortical development*” sounds like jumping to a conclusion too fast and before actually presenting the data).

Authors response: You have raised an important point here, thank you! We edited the sentence as follows:

“Comparing the levels of free mRNAs (input) with those bound to polysomes revealed that both transcriptional and translational control of gene expression are dynamically regulated during the development of the human neocortex (Fig. 1d, Table S3).”

→ similarly, in line 412 “*Data suggest that Celf4 expressing neurons are some of the earliest born neurons and serve an essential role in neurogenesis*” the statement is incorrect, as the data simply clarify the neurons’ birthdating, without providing any relevant information on their role in neurogenesis as their function has not been experimentally manipulated -which instead is what presented in the following paragraph.

Authors response: Thank you for pointing this out. As suggested by the Reviewer, we introduced the changes, and the new sentence reads as follows:

“These data suggest that Celf4+ neurons are among the earliest born and maturing neurons in the developing neocortex.”

The content of the paper could be organised in a much more systematic manner: jumping from human datasets to a comparison with mouse data makes it difficult to grasp the final message. Overall, the general organization and excessive length of the text tends to prevail, and this overshadows the genuine originality and strength of the excellent data.

Authors response: We are thankful that the Reviewer recognizes the genuine originality and strength of the excellent data presented in the manuscript. We regret a lack of clarity and better organization in the original manuscript and we now provide a substantially reorganized manuscript.

(3) The rest of the study explores the involvement of Celf4 in synaptic function and suggest that Celf4 might act as a translational repressor of synaptic mRNAs. This part of the study would be sufficient for a high-quality paper. It clearly demonstrates the importance of Celf4 availability for translation initiation and suggest that Celf4 deletion causes specific changes in translational control by favouring the shift of synapse-related mRNAs into the polysome-bound fractions.

Authors response: We thank the Reviewer for acknowledging the importance of our results. While we agree that the manuscript contains an extensive amount of data, the 1st co-author graduated and is moving to her new postdoctoral position at Yale University. Although it would be beneficial, extending the presented results into additional manuscripts is unfortunately not feasible.

(4) The sex differences are only analysed for a single gene. The authors found that non-nuclear VGAT+Gphn+ overlapping area was significantly increased in male mutants, while the opposite effect was observed in female mutants. Most of the description of the phenotype is at P0. It would be interesting to explore these differences at additional ages and extend the study to more genes.

Authors response: We appreciate the Reviewer’s insightful suggestion and agree that it would be useful to explore these differences at additional ages and extend the study to more genes. However, we believe that expanding our findings is neither feasible, given the costs involved and limited time that 1st co-author has to finalize this study. For this reason, we chose not to make this change, but we added the following sentence to the Discussion:

“Finally, and to our surprise, we found that Celf4 regulates prenatal GABAergic and glutamatergic synapses in a sex-specific fashion at P0 developmental stage, possibly contributing to the sex prevalence in a number of NDDs, including ASD. Future studies should examine if sex-dependent synaptic differences are present at other prenatal and postnatal stages, and whether Celf4 regulates the expression of other synaptic markers during neocortical development.”

(5) On several occasions, the authors fail to explain the reasons behind some of their choices and/or selections of data for specific analysis. For instance:

→ in the first paragraph, it is not clear why they did not check for the risk-gene set enrichment with an unbiased analysis across the entire snRNAseq dataset (i.e. all the clusters) instead of sub-setting the SP. Although it might be a relevant and understandable choice, without an explanation it gives the impression of a biased selection, especially considering that the authors themselves comment on the fact that the SP does not have enrichment for risk genes associated with severe forms of NDD.

Authors response: Thank you for this feedback; however, we would like to clarify that there might have been a misunderstanding regarding these analyses. As described in detail in the Methods subsection “Enrichment analyses”, we performed the risk-gene set enrichment analyses with unbiased approach using each single nucleus cluster from early fetal, early midfetal and late midfetal snRNAseq dataset. We have reworded the paragraph in main text (the third paragraph of section 1) as follows:

“To interrogate the relationship between neocortical development and risk for NDDs, we conducted risk-gene set enrichment analyses using permutation-based statistics across all clusters leveraging ASD risk genes emerging from exome sequencing analyses⁴¹ and genes associated with developmental delays curated by the Development Disorder

Genotype - Phenotype Database (DDG2P) ⁵² (Fig. 1b-c). For the DDG2P, we restricted to genes with brain involvement (see Methods). In agreement with previous reports ^{41,42}, we found an enrichment of ASD risk genes 41 in maturing and mature excitatory and inhibitory neurons, as well as in progenitors (Fig. 1b-c). Importantly, we found a striking enrichment of ASD risk genes (e.g., *SCN2A* and *GRIN2B* 35,42), but not DDG2P genes in SP clusters at both early fetal and midfetal periods (Fig. 1b-c; Table S2). The high expression of ASD genes in the SP is also corroborated by a published mouse dataset ⁵⁰. These data support previous evidence that the midfetal cortical development is a nexus of high vulnerability for ASD and NDDs ^{41,42,45-49} and further implicate SP neurons in the etiology of ASD.”

→ in the same section, the authors comment [lines 176-177] that a change in translation state might be driven by a change in cell-type composition, however, they do not clarify why they chose to do a bulk RNAseq considering that it might not be the best experimental choice in such instance.

Authors response: Thank you for this valid comment. In response to the Reviewer's comment, we revised the sentence in the main text as follows:

“We then conducted standard RNA sequencing to obtain information on the distribution of mRNAs along the polysome profile (GEO accession GSE214272). Standard RNA sequencing may not be the most suitable approach for studying translational changes that could be influenced by alterations in the cell-type composition during neocortical development; however, it remains a valuable and commonly used tool that can overcome the technical limitations of polysome fractionation, which requires cell lysis in a polysome-extraction buffer.”

→ they also explore the intersection of snRNAseq and polysome-associated changing transcripts at different stages, without specifying why they did not perform the analysis also with the monosome-associated transcripts.

Authors response: We appreciate this valuable comment. In response to the Reviewer's request, we conducted intersection analyses for the transcripts displaying changes in monosome fractions. The results of these analyses are depicted in new heatmaps, which have been added to Supplementary Figure 3. The main text and figure captions have been updated to reflect these additions.

Main text:

“By intersecting the snRNAseq datasets with the transcripts showing changes in monosome association, we identified 83 derepressed and 9 repressed mRNAs in early-to-mid transition, and 327 derepressed and 33 repressed mRNAs in early-to-late transition (Supplementary Fig. 3, left and middle, respectively). However, mid-to-late monosome comparison showed small level of translational changes with only 2 derepressed and 5 repressed mRNAs (Supplementary Fig. 3, right). Focusing on transcripts with changes in polysome association, we identified 73 derepressed and 16 repressed mRNAs in early-to-mid transition, 109 derepressed and 58 repressed mRNAs in early-to-late transition, and 3 derepressed and 14 repressed mRNAs in mid-to-late comparison. The mRNAs bound to different populations of ribosomes (monosomes and polysomes) in each snRNAseq cluster across the three developmental phases are listed in Table S5. ”

“In summary, these profiling results suggest that spatiotemporal control of neocortical translation is high during early-to-mid transition, and then declines from mid to late phases. Some translationally regulated transcripts display cell type-restricted expression (i.e. *CACNG8*, *GPC5*, *KCND3*, *NUCB2* in monosomes; *ZNF385B*, *CDKL2*, *ZFR2*, *COTL1* in polysomes). Most, though, lack cell specificity (i.e. *ANKIB1*, *KMT2E*, *LSAMP*, *NETO2*, *OGA* in monosomes; *LMO3*, *GNG2*, *LIMCH1*, *UCHL1* in polysomes).”

Figure caption:

“Supplementary figure 3. Translational changes of monosome-associated transcripts across neocortical development. Heatmaps showing the translational changes of monosome-associated mRNAs with comparable levels of expression in the input in each cell type at mid- (left) or late-phase (mid and right) snRNAseq clusters. The criteria used in Mono MvE (left), Mono LvE (middle), or Mono LvM (right) comparisons were $|\log_2FC| > 1$ and adjusted p-value < 0.05 .

The minimum mean expression threshold by snRNAseq clusters was set to > 0.1 with at least 30% of cells in a cluster expressing the gene.”

→ in the same paragraph, they do not explain why they chose to do the GO enrichment analysis only in the late cell clusters, instead of doing it also for the early ones.

Authors response: Thank you for this question. Following your suggestion, we conducted GO enrichment analysis for the transcripts that underwent translational regulation from the early to late developmental phase in the early clusters (SP&ExN L5/6 vs. ExN L5/6). In Figure 1g, we have replaced the GOCircles with the updated versions for both the subplate and deep layer clusters from the early and late phases. The modifications have also been made to the main text and figure captions, as described below:

Main text:

“Although monosomes are recognized as significant sites of active translation^{63,64}, our study focused on polysome-occupying mRNAs that synthesize multiple copies of a new protein from a single mRNA molecule^{65,66}, making them a canonical source of translation for dynamically evolving neocortical development^{23,24,54}. To determine the “Cellular Component” (CC) terms associated with both derepressed and repressed differentially expressed genes (DEGs) that change from early to late polysomes, we performed the GO enrichment analysis on early and late excitatory neuronal clusters (Table S5) using PANTHER Classification System version^{17,67}. While DEGs from early and late upper layer neuronal clusters did not yield any significant CC terms annotations, DEGs from SP and deep layer clusters showed substantial association with synaptic terms in both developmental groups. Specifically, we observed a significant enrichment of DEGs from the early “SP&ExN L5/6” clusters in two specific subclass terms highlighting “transmembrane reporter complex” and “synapse” (Fig. 1g, left, dark grey). Gene enrichments further progressed to “ionotropic glutamate receptor complex”, “integral component of postsynaptic density membrane”, “cation channel complex” and “glutamatergic synapses” in late “SP” clusters (Fig. 1g, right, dark grey). This suggests that fully developed SP tends to have greater expression of genes implicated in synaptic function. While DEGs from the early deep layer 5/6 neurons were primarily involved in “postsynaptic density membrane”, “glutamatergic synapses” and “transmembrane reporter complex” (Fig. 1g, left, light grey), the late excitatory 5/6 clusters were enriched for “mitochondrial respiratory chain complex I” and “integral component of postsynaptic density membrane”, with the latter showing higher fold enrichment compared to the early phase (Fig. 1g, right, light grey). The enriched GO-CC terms associated with synapses in both derepressed and repressed mRNAs (Fig. 1g) aligns with the notion that synapses first form in the SP and subsequently appear in layers 4-6 in developing mammalian neocortex^{4,6,7,9,15}. To further validate these results, we utilized the synapse biology SynGO database⁶⁸ (<https://syngoportal.org>) and confirmed that translationally-regulated mRNAs in both early and late SP and deep layer clusters (from Fig. 1g) are enriched for synaptic terms. SynGO revealed that 17 (out of 111) and 13 (out of 73) genes from the early “SP&ExN L5/6” and “ExN L5/6” clusters, respectively, were uniquely mapped to SynGO annotated genes, as compared to “brain” selected background (Table S6). Similar analyses were performed with DEGs from late “SP” and “ExN L5/6” clusters, where 22 out of 129 and 20 out of 125 genes were mapped to SynGO synaptic proteins, respectively (Table S6)”.

Figure caption:

“GOCircle plots show PANTHER enrichment analyses using translationally changed mRNAs in the (left) early “SP&ExN L5/6” (E0, E7, E9; dark grey) and “ExN L5/6” (E5; light grey) clusters, as well as (right) late “SP” (L4, L15, L16; dark grey) and “ExN L5/6” (L7, L10, L13, L14, L18; light grey). The gene list was obtained from “Poly LvE/early or late snRNAseq clusters” intersection analysis presented in Table S5. In GOCircle plot, the inner circle is a bar chart where the height of the bar represents the significance of the GO term ($FDR < 0.05$), and its color is linked to the z-score (the number of derepressed mRNAs minus the number of repressed mRNAs divided by the square root of the total number of counts). The outer circle denotes a scatter plot of the \log_2FC for each assigned derepressed (red dot) and repressed (blue dot) mRNA in each term. The tables below contain information (ID, term description and fold enrichment (F.E.) value) about all significant specific subclass cellular component terms (GO-CC), ranked by $FDR (< 0.05)$ significance.”

→ in section #7 the authors decide to have a closer look at upper-layer cellular subtype markers, without explaining the relevance of such cells at all – the entire focus of the paper is in subplate/deep-layer neurons considering that the protein of interest is expressed there, so it would have made more sense to look at those compartments more closely instead.

Authors response: We appreciate the Reviewer’s valuable suggestion. In our initial approach, we utilized upper layer clusters as a negative control to demonstrate that there is a significant enrichment of cellular component terms specifically in the SP and deep layer clusters. However, since there were no significant terms identified in the upper layer clusters, we have removed the corresponding GO Circles from Figure 1g. Nevertheless, we have incorporated a statement in the main text that summarizes our findings:

“While DEGs from early and late upper layer neuronal clusters did not yield any significant CC terms annotations, DEGs from SP and deep layer clusters showed substantial association with synaptic terms in both developmental groups.”

These are just few examples from the results, but there are many of these instances in the text, which brings it to the same conclusion: splitting the paper into different works might be helpful in increasing accuracy and efficiently to convey the compelling results described by the authors.

(6) In some instance, the results do not support the claims made by the authors. For example:

→ in paragraph #2 the authors conclude that a translational control mechanism might be responsible for the more rapid maturation of the human SP vs upper and lower neocortical layers potentially by redefining its synaptic protein repertoire from 1st to 2nd trimester. This contrasts with the choice of analysing only the late SP clusters, rather than looking also at the earlier stages.

Authors response: We appreciate your comment and have already incorporated your suggestion, as described under the point 5, arrow 4 of this document. Thank you for your valuable feedback.

→ in paragraph #7, the finding that the production, migration, and laminar organization of cortical neurons is not affected is in direct contrast with the previous comment regarding an “essential” role played by the molecule.

Authors response: Thank you for your kind reminder. As previously mentioned, we have removed the phrase “serve an essential role during neurogenesis.” The revised sentence now reads:

“These data suggest that Celf4+ neurons are among the earliest born and maturing neurons in the developing neocortex.”

→ in section #3 the authors search for candidate molecules expressed in both SP and L5/6, disregarding what they previously presented about the central role of the SP itself. It doesn’t make any sense to group together SP and L5/6, unless further explained and justified. The authors write: “*CELF4 can serve as novel and reliable marker of human SP and deep layers throughout neocortical development.*” If CELF4 is expressed in several cell populations than what is it marker of?

Authors response: Thank you for pointing this out. Following the revised differential gene expression analyses between pooled SP clusters and other pooled neuronal clusters for each developmental phase separately (as described

in authors response for Reviewer's question 3), we identified CELF4 as top-ranked RBP enriched for synaptic functions in the SP. It is important to note that the SP is still not fully differentiated during early fetal phase, and exist as deep presubplate layer (Kostović, 2020). For this reason, we identified clusters 0, 7 and 9 as "SP&ExN L5/6", while distinct SP clusters are detectable only during early midfetal and late midfetal phases (as shown in Fig. 1a). Furthermore, CELF4/Celf4 protein expression is most prominent in the SP area but is also detected in layers 5/6 in both human (Fig. 2b, Supplementary fig. 4a) and mouse (Fig. 3a, Supplementary fig. 5) developing neocortices. Similarly, CTIP2/Ctip2 and TLE4/Tle4 have been promulgated as selective markers of deep layer neurons and the SP, with a widely accepted notion that CTIP2/Ctip2 predominantly labels layers 5 and TLE4/Tle4 layer 6 neurons. Hence, we believe that CELF4/Celf4 is indeed a marker of the SP and deep layers, with a more prominent expression in the SP neuronal subtypes.

The Salamon et al., manuscript contains highly interesting and valuable data and I am supportive in its publication. Nevertheless, I have to say that the paper contains too much data that are not all necessary to present together. The manuscript is hard to follow, and it is not an easy read. I urge the authors to streamline the story and consider presenting the data in separate paper. They could consider 2-3 excellent paper with clear and concise messages.

We are grateful for the support from this Reviewer. Our aim was to demonstrate the integrative nature of prenatal phases in neocortical development, highlighting their correlation in both space and time with other developmental processes occurring during pioneering synaptogenesis. The latter serves a dual role, contributing to both morphogenetic changes and temporary functionality. It is important to note that primary synaptogenesis takes place after birth.

References:

Kostović, I. (2020). The enigmatic fetal subplate compartment forms an early tangential cortical nexus and provides the framework for construction of cortical connectivity. *Progress in Neurobiology* 194, 101883. doi: 10.1016/j.pneurobio.2020.101883.

Reviewer 3 (Remarks to the Author):

Reviewer 3 Comments:

(1): Fig.1a, Supplementary Fig.1_All six dataset should be integrated and processed for the downstream filtering and clustering analysis (Seurat provides such a framework).

Authors response: We understand the Reviewer's viewpoint here and appreciate your suggestion. We had originally integrated all the data as the Reviewer suggested but chose to display each time point clustered separately for clarity. The phase-specific clustering correctly partitions cells by discernable cell types based on the highly dynamic phases of cortical development (Supplementary Fig. 1a, bottom), which is important for our interpretation. In contrast, integrating all the data together was not ideal for our analysis since, even after optimal clustering, some populations of nuclei were not well-separated into distinct clusters (Supplementary Fig. 1a, top), but instead were spread across multiple phase-specific clusters (heatmaps in Supplementary Fig. 1b). This approach would have missed important molecular mechanisms that distinguish between different biologically relevant states. Given that we did not directly examine changes in clusters over time, working off the same UMAP projection and clustering was not important for our strategy. Also, we updated the color scheme of the original UMAP plots in Fig. 1a to make distinct clusters easier to see. We also introduced the changes in the main text, figure captions and material and methods; the new sentences read as follows:

Main text: "Rather than integrating all the data into one analysis, we opted for phase-specific integration and clustering. This allowed us to accurately group nuclei into clusters matching discernible cell types, providing a more precise understanding of the developmental process. Using Uniform Manifold Approximation and Projection (UMAP) to visualize the shared nearest neighbor (SNN) clustering of the phase-specific datasets, we identified 19 early, 21 mid, and 21 late clusters (Supplementary Fig. 1a, bottom). In contrast, integrating all samples followed by optimal clustering led to the formation of clusters that do not clearly match individual cell types (Supplementary Fig. 1a, top) since some nuclei from phase-specific clusters were allocated to different integrated clusters, potentially rendering them less biologically interpretable (Supplementary Fig. 1b). This suggests that the integration approach is confounded by similar cells from other phases, therefore differing from the phase-specific clustering approach, which highlights the importance of carefully considering the most appropriate analysis strategy for the specific research question at hand."

Figure captions: "Supplementary figure 1. UMAP visualization of integrated or phase-specific clustering of nuclei from human neocortical samples across three developmental phases of mid-gestation.

(a) UMAP projections, split by phase, showing clustering of the integrated samples identified 29 clusters (top), while developmental phase-specific clustering identified 19 early-, 21- mid and 21 late-specific clusters (n = 2 neocortices per each developmental phase). Nuclei within each developmental group (early, mid, or late) are colored by SNN cluster assignment (using Seurat FindClusters). (b) Heatmaps showing the numbers of cells from each phase-specific cluster intersecting clusters of the integrated analysis. For example, early cluster 0 from phase-specific analysis includes the cells from combined early clusters 5, 6, and 8. Nuclei from the combined early cluster 5 can be found in early phase-specific clusters 0, 4, 5, 6, 7, and 9."

Material and methods: "Using the single cell R package Seurat (version 4) 90, the matrices from individual samples were loaded into a single Seurat object, clustered by a shared nearest neighbor (SNN) algorithm, colored by SNN cluster assignment using Seurat FindClusters and visualized per cell type in the two-dimensional space using scCustomize v.1.1.1 (<https://doi.org/10.5281/zenodo.5706430>; ref. ⁹¹)."

(2): Fig.1a and line 136-_To objectively assess developmental changes in snRNA clusters, the authors need to perform pseudotime analysis e.g. using Monocle or Slingshot.

Authors response: We understand that pseudotime analysis can be helpful for many applications. However, since we used the snRNAseq data to isolate distinct, phase-specific clusters corresponding to SP and its components, we do not believe it would add useful information for our purposes. Our goal was not to trace changes over developmental phases directly so adding a pseudotime analysis would only be distracting.

(3): Fig.2a The discovery process of CELF4 should be more objective. One approach would be to perform a differential expression analysis between SP and other neuronal clusters. The authors should provide statistics for unbiased evaluation of the SP-specific gene(s) (Seurat provides such a framework).

Authors response: This is a very good suggestion, which we have adopted in the revised manuscript. In brief, we conducted separate differential gene expression analysis between pooled SP clusters and other pooled, neuronal clusters for each developmental phase. Then, we performed gene ontology analysis to identify which DEGs were associated with the greatest numbers of biological process terms associated with synaptic function. Lastly, we intersected our DEG lists with the RBP2GO list, consisting of 6100 human RBP candidates, to identify RBPs with the highest RBP2GO score. The unbiased summary statistics are presented in a new Table S7. The main text and figure captions have been modified accordingly:

Main text: “Seeking to identify genes that are expressed in both SP and layer 5/6 neurons, and can potentially regulate the translation of synaptic genes, we narrowed the scope of our investigation to RBPs, which are the major regulators of mRNA translation^{16–19}. To identify candidate RBPs enriched in SP and associated with predicted synaptic function, we conducted DEG analyses between SP clusters and other neuronal clusters (Fig. 2a), separately for each developmental phase. We then filtered the identified DEGs by mapping them to GO terms associated with synaptic function and counted the number of terms associated with each gene. Moreover, we compared the DEG lists for each developmental phase with a collection of 6,100 human RNA-binding protein (RBP) candidates from the RBP2GO database⁶⁹ to determine which RBPs were significantly upregulated in the SP clusters. After ranking the RBPs by their RBP2GO scores, we found that CUGBP Elav-Like Family Member 4 (CELF4) was the top-ranked RBP in all three developmental phases (Table S7).”

Figure captions: “(a) Venn diagrams showing all RBPs that were significantly upregulated in the SP clusters versus all other neuronal clusters after pre-filtering Table S7 for DEGs that (1) are present in the RBP2GO list, (2) have $\text{avg_log2FC} > 0$ (up-regulated), (3) have one or more GO-BP terms associated with synaptic function, (4) have RBP2GO scores > 0 . CELF4 was the top-ranked RBP in all three developmental phases. The DEG analysis was conducted by comparing SP clusters against all other neuronal cluster, individually for early (SP: E0, E7, E9 versus Neuronal: E2, E4, E5, E6, E8, E16), mid (SP: M5, M13 versus Neuronal: M0, M1, M2, M3, M7, M8, M9, M10, M11, M12) and late (SP: L4, L16, L15 versus Neuronal: L0, L1, L2, L3, L5, L6, L7, L10, L13, L14, L18).”

(4): Fig.2h Co-expression of CELF4 and its candidate target mRNAs at single cell level should be examined more objectively (by using e.g. COTAN <https://academic.oup.com/nargab/article/3/3/lqab072/6348150>). The representation in the provided figure is not convincing enough to draw conclusions.

Authors response: Thank you for your valuable suggestion. We agree that COTAN is indeed innovative approach and can provide direct insights into the co-expression of gene-pairs of interest. Our hypothesis was not that these two genes were coordinately expressed, but that one was available in a preponderance of cells co-regulate with the other. COTAN was not helpful for testing this since it focused on expression levels. However, we decided to illustrate our concept by replacing the originally submitted Fig. 2h with a stacked percentage bar plot showing the actual number of nuclei within each cluster expressing *SV2A* alone or in combination with *CELF4*, presented as the percentage of all

nuclei in each cluster on the y-axis. We believe that this visualization highlights the cellular co-expression patterns of these two transcripts in our dataset. We also modified the main text and figure captions to reflect the changes made.

Main text: “Therefore, we sought to examine the percentage of nuclei expressing *SV2A* mRNA alone or in combination with *CELF4* from the total number of nuclei in each cluster across early, mid and late phases of neocortical development (Fig. 2h). Although *SV2A* mRNA was detected in all nuclei clusters, the percentage of nuclei expressing *SV2A* varied across developmental phases. Notably, excitatory neuronal clusters consistently displayed a high number of *SV2A*-expressing cells, while the non-neuronal clusters displayed the opposite trend. Out of total number of nuclei with detectable *SV2A* expression, the SP and excitatory layer 5/6 clusters showed relatively high co-expression of *SV2A* and *CELF4*, ranging from 81% to 59% in the early phase, 75% to 53% in the mid phase and 83% to 60% in the late phase (Fig. 2h). Given that both transcripts are frequently co-expressed in the same cell, it is plausible that the role of RBP *CELF4* and *SV2A* could converge on synaptic functions.”

Figure captions: “Stacked bar plot illustrates the percentage of nuclei expressing *SV2A* alone (pale turquoise) or in combination with *CELF4* (tan, rose, slate blue), expressed as percent of all nuclei in each cluster across early, mid, and late phases of cortical development. The actual nuclei count values are labeled within each bar for each cluster.”

Other minor concerns are as follows:

(5): **Fig.1f_I** find a conceptual contradiction in intersecting snRNAseq data and polysomal RNAseq data, since they are derived from different subcellular origins; the former is from the nucleus and the latter is from cytosol. Previous reports support that the RNA levels are mostly concordant between nucleus and cytosol (e.g. <https://www.nature.com/articles/s41598-017-04426-w>) and I would add such a paragraph mentioning the authors performed this analysis based on the assumption that the RNA levels are concordant between nucleus and cytosol.

Authors response: You have raised an important point here, thank you! To facilitate interpretation of Fig. 1f and 1g, we accepted your suggestion and included the following sentence in the second paragraph of the results section *Dissecting the developmental translational dynamics in human fetal cell types*:

“This direct comparative analysis was based on the assumption that single nucleus transcriptome profiles recapitulate faithfully many of the transcriptomic changes found in intact single cells^{62,63}, as carefully demonstrated by other side-by-side studies showing high correlation between single nucleus and whole cell transcriptomes⁶⁴⁻⁶⁶.”

(6) **lines 125-126:** “*bulk transcriptomic data*” should be corrected to “single nuclei transcriptomic data” or something.

Authors response: We apologize for this error. We removed the “bulk transcriptomic data” from the modified sentence (please check question below).

(7) **lines 174-175:** “*Comparing the levels of free mRNAs (input) with those bound to polysomes revealed that translational regulation couples with transcription to define human neocortical development*” This statement sounds too definitive. While this finding suggests the importance of translation control in human neocortical development, it may be misleading to say that it defines human neocortical development.

Authors response: We fully agree with the Reviewer that this statement was too strong. We accepted your suggestion and new sentence reads as follows:

“Comparing the levels of free mRNAs (input) with those bound to polysomes revealed that both transcriptional and translational control of gene expression are dynamically regulated during the development of the human neocortex (Fig. 1d, Table S3).”

(8) line 200: “*expression cutoff frequency to 30% of cells within a cluster*” should be missing the cutoff criteria for log2FC increase threshold (default setting by Seurat is 0.25 but the method describes “1.2-fold cutoff” so please clarify.

Authors response: Thank you for this comment. Our goal was to eliminate differential expression data resulting from genes expressed in relatively few cells within a cluster. We selected an expression cut-off because a similar filter had been effectively used in a prior study (Polioudakis et al., 2019). We tested a range of frequencies and found that 30% provided consistent differences while avoiding clusters with few cells having detectable expression. Thank you for bringing our attention to the mistake in the Material and Methods subsection “Single-nucleus RNA sequencing (snRNAseq)”. We apologize for any confusion this may have caused and appreciate your help in identifying the error. Upon reviewing our methods, we realized that we had inadvertently reported the wrong criteria for snRNAseq (sentence: “*Differential expression used a 1.2-fold cutoff with at least 30% of cells in a cluster expressing the gene.*”). We have corrected the sentence to accurately reflect the correct criteria as follows:

Material and methods: “For intersection analyses, we used minimum mean expression by cluster (> 0.1) with at least 30% of cells in a cluster expressing the gene.”

Main text updated: “To get a snapshot of the complex translational status of each cell type across human developmental periods, we compared our polysome profiling datasets with our snRNAseq screens. Therefore, we considered mRNAs with $|\log_2FC| > 1$ in the polysome profiling data, > 0.1 minimum mean expression by snRNAseq cluster threshold and at least 30% of cells in a cluster expressing the mRNA.”

Figure captions: “...after using minimum mean expression by cluster (> 0.1) with at least 30% of cells in a cluster expressing the gene.”

(9) There are some typos or sentences that are a bit hard to interpret. Proofreading by a trained yet unbiased staff member is recommended.

Authors response: We regret there were problems with the English. We have given the manuscript a further proofread and asked a native English speaker to carefully revise the manuscript to improve the grammar and readability. We hope that all typos and ambiguous sentences have been eliminated.

(10) I would request the authors to describe more on the phenotype of the Celf4-cKO mouse. I would discuss how the “*recurrent seizures later on*” (line 698) is comparable to the previous findings on Celf4 KO mouse (e.g. <https://www.sciencedirect.com/science/article/abs/pii/S0006291X22003989>) and relate it to human neurodevelopmental disorders. I would also request to mention whether there is a sex difference in the occurrence of seizures in Celf4-cKO mouse.

Authors response: Thank you for these comments. We observed occasional hypo- and hyperactivity and spontaneous behavioral seizures only during postnatal stages and adulthood in both male and female *Emx1-Cre Celf4* mutants. These findings are subject of future studies and are outside the scope of this study. Nevertheless, during a course of ECT treatment, (Wagnon et al., 2011) also showed that the seizure threshold of female and male *Emx1-Cre Celf4* cKOs was significantly reduced when compared to the sex-specific controls. The same study also revealed that female WTs have a lower standard threshold than the male WTs, supporting our findings of the sex-dependent synaptic

phenotype upon *Celf4* deletion from excitatory neurons (presented in Fig. 5). As suggested by our Gene Ontology enrichment analysis (Fig. 3g) and our RIP-RNA seq data (Figs. 2f and 3c), it is thus possible that loss of *Celf4* from excitatory neurons leads to various synaptic defects that can consequently impair neuronal/ cortical excitability and lead to the numerous neurodevelopmental disorders, such as epilepsy and depression. For this reason and inspired by the publication (<https://www.sciencedirect.com/science/article/abs/pii/S0006291X22003989>) suggested by the Reviewer, we included our response to paragraph 7 in the discussion:

Consistent with the synaptic deficit phenotype (Fig. 5) and previously published data^{77,78}, we observed that *Celf4* loss causes occasional hypo- and hyperactivity and spontaneous recurrent seizures later on, during mouse postnatal and adult stages (not shown). Our GO enrichment analyses (Fig. 3g) and RIP-RNA seq data (Figs. 2f and 3c) suggest that early prenatal *Celf4* loss from excitatory neurons may result in various synaptic defects in the SP, potentially impairing cortical excitability postnatally and causing epilepsy and depression^{77,78,91}.

References:

- Polioudakis, D., de la Torre-Ubieta, L., Langerman, J., Elkins, A. G., Shi, X., Stein, J. L., et al. (2019). A Single-Cell Transcriptomic Atlas of Human Neocortical Development during Mid-gestation. *Neuron* 103, 785-801.e8. doi: 10.1016/j.neuron.2019.06.011.
- Wagnon, J. L., Mahaffey, C. L., Sun, W., Yang, Y., Chao, H.-T., and Frankel, W. N. (2011). Etiology of a genetically complex seizure disorder in *Celf4* mutant mice. *Genes Brain Behav* 10, 765–777. doi: 10.1111/j.1601-183X.2011.00717.x.

REVIEWERS' COMMENTS

Reviewer #1 (Remarks to the Author):

the authors have addressed my comments.

Reviewer #2 (Remarks to the Author):

The authors considered all my suggestions and revised the manuscript to my satisfaction. I also appreciate that they made an effort to streamline a complicated story and increased clarity. I have no further criticisms.

Reviewer #3 (Remarks to the Author):

The authors made diligent efforts to address the concerns raised, and I wholeheartedly support the validity of the data and contents, which convincingly meet the standards for acceptance to Nature Communications.